# A non-canonical activation of the host's ESCRT machinery is required for the scission of parasitophorous vacuoles and the replication of *Leishmania donovani*

Javier Rosero, Peter E. Kima ⓘ*

Department of Microbiology and Cell Science, Gainesville, Florida, United States of America

* pkima@ufl.edu

## Abstract

*Leishmania donovani* (Ld) is the causative agent of visceral leishmaniasis, which results in death if not treated. In mammalian cells, Ld live in vacuolar compartments called *Leishmania* parasitophorous vacuoles (LdLPVs) that enigmatically divide following parasite replication. We evaluated the role of the endosomal sorting complex required for transport (ESCRT) machinery in the scission of LdLPVs. We found that ESCRT components are constitutively recruited to LdLPVs. We propose that this recruitment depends on the expression of PI(3,4)P2 on LdLPVs. The knockdown (KD) of upstream components of the ESCRT machinery revealed that ALIX, but not TSG101 or VPS28, led to a significant reduction in the parasite burden in infected cultures. Interestingly, LdLPVs in ALIXKDs were more distended and harbored more than 2 parasites. Incorporation of BrdU into *Leishmania* in THP-1 macrophages revealed that parasite replication was inhibited in ALIXKD due to defective LdLPV scission. These findings establish that non-canonical activation of the ESCRT machinery is required for *Leishmania* to replicate within macrophages.

## Author summary

The endosomal sorting complex required for transport (ESCRT) machinery plays critical roles in physiological processes, including the biogenesis of extracellular vesicles within multivesicular bodies and cytokinesis. In addition, it can be hijacked to promote the spread and persistence of infectious agents, including the budding of viruses and the acquisition of nutrients by pathogens. In this study, we uncover a new role for the ESCRT machinery in the infection of macrophages by *Leishmania donovani*. Within mammalian cells, LdLPV that harbor individual *Leishmania donovani* parasites enigmatically divide to accommodate daughter parasites. We demonstrate that the engagement of the ESCRT machinery,

**Data availability statement:** All relevant data are within the manuscript and its Supporting information files.

**Funding:** The author(s) received no specific funding for this work.

**Competing interests:** The authors have declared that no competing interests exist.

initiated by the recruitment of ALIX, an ESCRT accessory molecule, is required for the division of LdLPVs and parasite persistence. Interestingly, the knockdown of other upstream ESCRT components, including TSG101 and VPS28, had no discernible impact on LdLPV division. Future studies on the mechanisms of selective ESCRT machinery activation initiated by ALIX recruitment are expected to reveal targets for controlling this deadly pathogen.

## Introduction

*Leishmania* spp. parasites are single-celled protozoan organisms that are transmitted by sandflies. *Leishmania donovani* is the causative agent of visceral leishmaniasis, which has an estimated 50,000–90,000 new cases per year, 95% of which are fatal if left untreated [1]. Visceral leishmaniasis presents as irregular bouts of fever, weight loss, splenic and hepatic enlargement, and anemia [2]. In addition to infecting monocytes and macrophages, *L. donovani* infects hematopoietic stem cells, megakaryocytes, and other cell lineages that are not known to be phagocytic [3]. Once inside mammalian cells, *Leishmania* spp. are lodged in membrane-enclosed compartments, often referred to as *Leishmania* parasitophorous vacuoles (LPVs). LPVs fuse extensively with late endosomal compartments and with vesicles from the secretory pathway [4–6]. Interestingly, LPVs that harbor *L. donovani* (LdLPVs) retain early endosomal characteristics over a prolonged period as they acquire late endosomal characteristics slowly [7]. Moreover, LdLPVs harbor individual parasites and enigmatically divide following parasite replication to accommodate daughter parasites that segregate into separate LdLPVs [4,8]. *Leishmania* replication in mammalian cells is poorly understood. It is known that soon after the internalization of the infective metacyclic promastigote form into mammalian cells, *Leishmania* parasites commence an orderly transformation process characterized by the gradual shutdown of promastigote-specific molecules and the synthesis of intracellular stage molecules, including A2 [9,10]. Approximately 24 hours after the parasite enters cells, and depending on the type of cell, parasite replication commences [9]. In infections with *L. donovani* where parasites live in separate LPVs, parasite division is enigmatically linked to the scission of the LPV. As we considered the likely underlying mechanisms of LdLPV scission or division, we wondered whether it is mechanistically like the division of organelles within the cell, including the division of mitochondria [11,12]. Alternatively, the LdLPV division could be mechanistically like the budding of enveloped viruses from membrane-delimited compartments, including the nuclear or plasma membrane, as described in studies on the egress of EBV from the nucleus or HIV from the plasma membrane of infected cells [13,14]. It could also be mechanistically like cytokinesis, where a cytoplasmic bridge at the midbody between nascent daughters is eventually cleaved in a process called abscission [15,16]. In both cytokinesis and viral budding, the endosomal sorting complex required for transport (ESCRT) machinery has been implicated.

The ESCRT machinery is composed of 4 molecular complexes that are functionally conserved in eukaryotes. In mammalian cells, ESCRT 0 members include the hepatocyte growth factor-regulated tyrosine kinase substrate (HGRS-1) and signal transducing adaptor molecule (STAM) 1 and 2 [17]. The ESCRT I complex includes tumor-susceptibility gene 101 (TSG101), vacuolar protein sorting (VPS) members VPS28, VPS37 (A-D), and UBAP1 [18]. ESCRT II includes subunits of ELL-associated protein 20/vacuolar protein sorting (EAP) (20, 30, and 45) [19]. While ESCRT III includes the Charged membrane proteins (CHMPs) 1, 2, 3, 4, 6, and 7, with 2 variants for CHMP1 (a and b), 2 variants for CHMP2 (a and b), and 3 variants of CHMP4 (a, b, and c) [20]. In addition to these complexes, there are accessory molecules that associate with some complexes and that play crucial roles in the overall functions of the ESCRT machinery. Notably, ALG-2 interacting protein X (ALIX) is an accessory molecule to the ESCRT I complex and may substitute for ESCRT I in some activities [21]. The AAA ATPase vacuolar protein sorting-associated protein (VPS4 (a and b)) is another accessory molecule that interacts most extensively with members of the ESCRT III complex and catalyzes membrane scission by ESCRT III and promotes their recycling from membranes [22]. Activation of the ESCRT machinery for the formation of intraluminal vesicles within multivesicular endosomes, for example, follows a canonical scheme where there is sequential recruitment of ESCRT complexes [23,24]. Increasingly, studies on the function of the ESCRT machinery are revealing the non-canonical participation of ESCRT complexes [21,25,26], where some complexes appear to be dispensable to accomplish tasks of the ESCRT machinery. For example, although the role of ALIX in cytokinesis remains unsettled, some studies have found it to be more essential in this process [27–29]. Similar observations were made in studies of plasma membrane repair [30] and HIV budding from the plasma membrane [14]. Such non-canonical schemes for activating the ESCRT machinery have led to approaches that selectively target specific ESCRT molecules to limit well-defined ESCRT-dependent processes [25,31,32]. Conversely, the occurrence of lethal diseases due to defects in specific ESCRT members highlights the crucial roles that the ESCRT machinery plays in maintaining good health [33].

The ESCRT machinery has been implicated in the pathogenesis of intracellular bacteria and parasites that reside within vacuolar compartments in infected cells [34]. Most intracellular bacteria express secretion systems whose components are inserted into the vacuole-delimiting membrane. Insertion of the secretion apparatus into the vacuole membrane causes some damage, which then activates the ESCRT machinery that responds to any damage in the endomembrane systems [35,36]. Such is the case with *Mycobacteria* infections, where damage of the *Mycobacterium-containing* compartment (MCV) has been shown to occur due to the insertion of ESX machinery, components of the Type VII secretion system [37,38]. Damage to the MCV is known to result in the spillage of *Mycobacteria-derived* molecules, including nucleic acids, to the cytosol, where they activate the cytosolic surveillance response that induces innate immune responses. The ESCRT machinery is then activated to repair the damaged MCV endomembrane [38–41]. In *Salmonella* infections of epithelial cells, where bacteria reside in the *Salmonella-containing* compartment (SCV), which is an intricate network of tubules, the bacteria express the Type III secretion apparatus that is deployed to release effectors into the cell cytoplasm. It was shown that components of the ESCRT machinery are recruited to the SCV, presumably to limit damage from the secretion apparatus [42]. In *Coxiella burnetii* infections, too, damage to their pathogen-containing vacuole was suggested by the recruitment of galectin 3, a known indicator of lysosome damage [43]. This was believed to be the trigger for the recruitment of ESCRT components.

Membrane damage is not the only inducer for the recruitment of ESCRT molecules to pathogen-containing vacuolar compartments. In *Toxoplasma* infections, where the parasite-containing vacuoles (TgPV) are non-fusogenic, it was found that components of the ESCRT machinery are recruited to the TgPV membrane (TgPVM) [44,45]. There, they interact with parasite-derived molecules, including the GRA proteins that are inserted into the TgPVM. As regards the function of the components of the ESCRT machinery that are recruited to the TgPVM, they were implicated in the acquisition of nutrients from the host cell [45]. This function of the ESCRT machinery is different from the canonical functions of the ESCRT machinery in cytokinesis and extracellular vesicle biogenesis [36]. It is not known what the underlying mechanisms are that preferentially activate the ESCRT machinery to accomplish such non-canonical functions. In the study by

Riviera-Cueves and colleagues [45] the interactions of TSG101 with the TgPVM were found to be more consequential for the uptake of nutrients than the interactions of ALIX with parasite molecules at the TgPVM.

There is evidence that some phosphoinositide species are the membrane anchor to which ESCRT components are recruited [46–48]. It was shown that for the completion of cytokinesis, phosphoinositides serve as critical membrane anchors for ESCRT molecules. In the lens of the eye, for example, the $PI(3,4)P_2$ (phosphatidylinositol 3,4-bisphosphate), was identified as the binding partner of the ESCRT II complex member VPS36 [47]. Interestingly, the absence or loss of either the lipid or the complex member resulted in impaired cytokinesis.

As discussed earlier, *Leishmania* resides in fusogenic endocytic compartments. Unlike intracellular bacteria, *Leishmania* lack a secretion apparatus that delivers pathogen-derived molecules across the LPV to the cell cytosol. It is, therefore, not known whether, in the absence of vacuole damage-inducing machinery, there are other inducers of ESCRT recruitment to the LdLPVs. The studies here were initiated when we sought to investigate the biogenesis of extracellular vesicles (EV) released from infected macrophages [49]. Proteomic analysis of EVs derived from infected cells had shown significantly higher levels of ESCRT components as compared to EVs from non-infected cells [49]. Preliminary studies then revealed that members of the ESCRT I and ESCRT III complexes are recruited to LdLPVs. We then proceeded to track the recruitment of representative ESCRT molecules to LdLPVs in *L. donovani*-infected cells. We took advantage of the availability of the dominant negative construct of VPS4 (pEGFP-VPS4-E228Q) whose expression has been shown to slow down the recycling of ESCRT III from membranes [45,50], which therefore permits the visualization of transient associations of the ESCRT machinery with membranes. To gain insight into how cells would respond to damage of LdLPVs, we studied the effects of the lysosome membrane-rupturing agent l-leucyl-l-leucine methyl ester (LLOME) on the recruitment of ESCRT molecules to LdLPVs. To begin addressing the critical roles of an activated ESCRT machinery in *Leishmania*-infected cells, we monitored the effect of knocking down TSG101 or ALIX. TSG101 and ALIX can function in the recruitment of downstream components, including ESCRT III members [26]. The effect of ALIX knockdown was dramatic. There was an increase in the frequency of larger LdLPVs harboring greater than 4 parasites. The LdLPVs division was defective in ALIX knockdowns due to the failure of proper ESCRT machinery activation. BrdU incorporation studies, which were more interpretable in the THP-1 cell line, confirmed the essentiality of ALIX for LdLPV division, specifically and more globally, to *L. dononani* parasite replication.

## Results

### ESCRT I and ESCRT III molecules are recruited to LdLPVs

All members of the ESCRT machinery are cytosolic proteins that are recruited to membranes where they exercise their functional activities [25,35]. We wanted to know whether the ESCRT machinery is assembled on LdLPVs. To supplement observations using commercially available antibodies to mouse ESCRT molecules, we elected to evaluate the redistribution of fluorophore-tagged recombinants of a representative ESCRT I member, TSG101 (mEGFP-TSG101), representative ESCRT III members, CHMP2B (mCherry-CHMP2B) and CHMP4B (mCherry-CHMP4B), in *L. donovani*-infected RWA264.7 macrophages. We also evaluated the distribution in infected cells of the ATPase VPS4A (mCherry-VPS4A), an accessory molecule critical for recycling ESCRT III molecules from membranes. The experimental scheme is illustrated in Fig 1A. On average, just under 60% of macrophages were infected under our experimental infection conditions. Unlike in uninfected cells, where these molecules are diffused and punctately labeled, in some infected cells, they are recruited to and displayed on LdLPVs. Representative images showing the distribution of the tagged molecules in uninfected cells and their recruitment to LdLPVs in infected cells are presented in Fig 1B, Fig 1C. Parasite nuclei and host cell nuclei were detected with DAPI staining. To visualize the contours of the LdLPV membrane, the lysosome-associated membrane protein 1 (LAMP1) was labeled. In some infected cells, the tagged ESCRT molecule was at the midpoint of two LdLPVs (Fig 1B, Fig 1C).

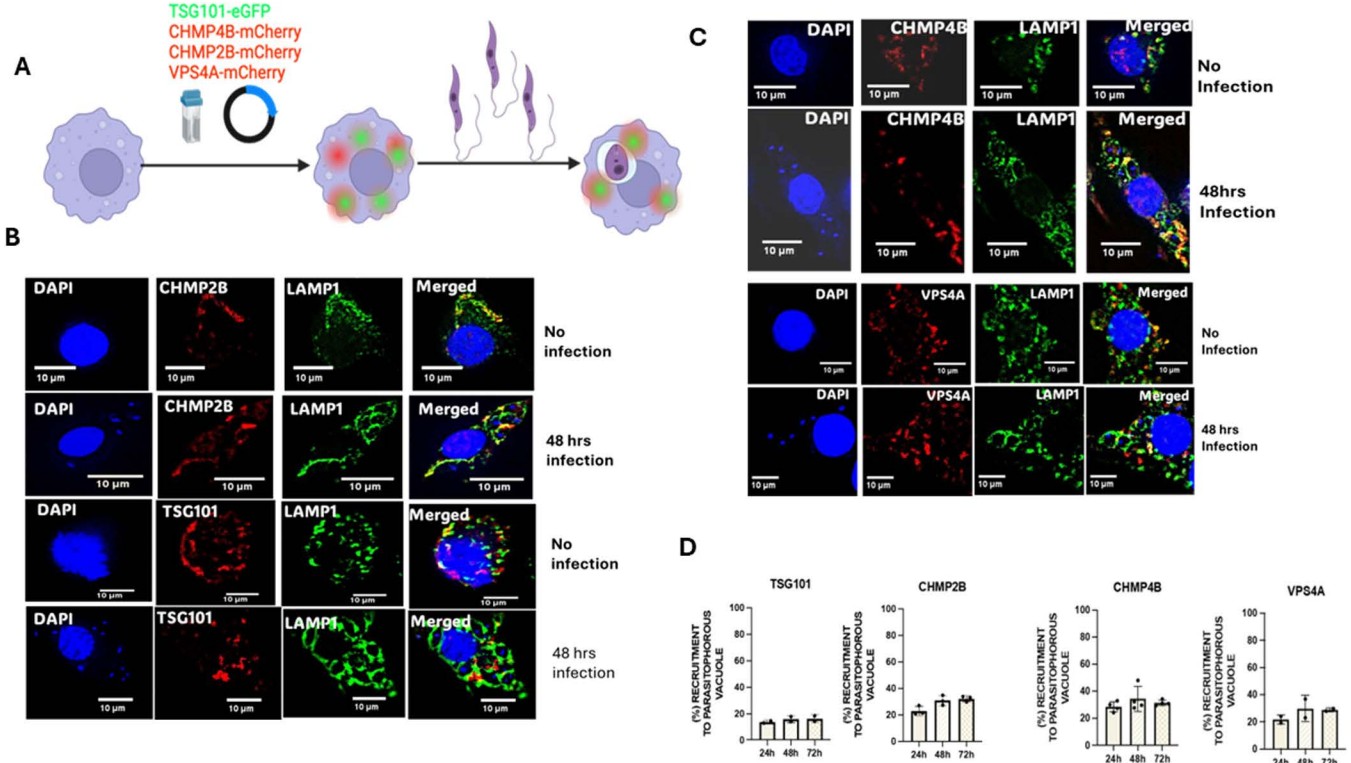

**Fig 1. The ESCRT machinery assembles on LdLPVs.** (A).The scheme for expressing recombinant fluorophore-tagged components of the ESCRT machinery in RAW264.7 macrophages is shown. Transfected cells on coverslips were infected for 24, 48 or 72 hrs with metacyclic *L. donovani* parasites. B) Representative images of transfected and infected cells are shown. Parasite and host cell nuclei are labeled with DAPI (blue). C) The proportion of *Leishmania* parasitophorous vacuoles with colocalization of the fluorophore-tagged iESCRT molecule was plotted. At least 100 parasite vacuoles were counted per coverslip time point. Counts were performed in duplicate on coverslips for at least three independent experiments. One-way ANOVA with multiple comparisons followed by *post hoc* Tukey's honest significant difference test to determine statistical significance was performed. *p < 0.05, **p < 0.01, ***p < 0.001, ****p < 0.0001. N. S, not significant.

We proceeded to count the number of LdLPVs within transfected cells that were deemed to be positive for each of these molecules. The counts compiled from 3 experiments are shown in Fig 1C. Approximately 15% to 19% of LdLPVs displayed TSG101-GFP at 24, 48 and 72 hours after infection. Approximately 21% to 26% of LdLPVs positively displayed CHMP2B-mCherry or CHMP4B-mCherry at 24,48 and 72 hours after infection. A comparable number of LdLPVs were positive for VPS4A-mCherry at the exact times post-infection.

Inert particles, including latex beads or *Zymosan* particles that are internalized by phagocytosis, are often used to probe the interactions of phagocytic compartments as they traverse through the endocytic pathway [51,52]. Here, we monitored the recruitment of CHMP2B and CHMP4B to phagosomes harboring 2 micrometer-sized latex beads. Representative images from CHMP2B-mCherry transfected cells harboring latex beads show no recruitment to the phagosomes (S1 Fig). The enumeration of phagosomes harboring latex beads shows that less than 5% of those vacuoles displayed CHMP2B-mCherry or CHMP4B-mCherry molecules. Our interpretation, therefore, is that the composition and characteristics of LdLPVs that permit the recruitment of ESCRT molecules differ from those of generic phagosomes.

Currently, there is no information available that provides insight into the expected frequency of ESCRT component recruitment to LdLPVs [45] and others [50,53] had reported on the value of expressing VSP4-E228Q, a dominant negative variant of VPS4A in cells. The inability of this variant to hydrolyze ATP prevents the recycling of ESCRT III complexes from

membranes, thereby forcing their accumulation. We proceeded to perform a dual transfection of pEGFP-VSP4-E228Q and CHMPB-mCherry following the scheme in Fig 2A, followed by infection with *L. donovani* (Fig 2). Representative images of transfected cells without infection and transfected cells with infection are shown (Fig 2B). The figure shows a more accentuated labeling of CHMP4B-mCherry on LdLPVs. Enumeration of the number of LdLPVs that were positive for CHPMP4B-mCherry revealed that greater than 90% were positive for this molecule (Fig 2C). We also determined that the course of infection in cells transfected with pEGFP-VSP4-E228Q was comparable to the infection in control cells (Fig 2D). When the recycling of ESCRT III molecules was prevented due to the expression of VSP4-E228Q, there was an accumulation of CHMP4B on most LdLPVs. This result suggested that the counts for the proportion of LdLPVs with recruited ESCRT molecules presented in Fig 1C were possibly an undercount of the number of LdLPVs that recruit CHMP4B-mCherry. Taken together, these results suggest that ESCRT molecules are constitutively recruited to LdLPVs. However, a subsequent signal may be required to trigger the nucleation of some ESCRT components and the aggregation of the ESCRT machinery, resulting in more pronounced labeling of these components.

### Effect of LLOMe on ESCRT recruitment to LdLPVs

Several studies have shown that damage to lysosomes is a potent activator of the ESCRT machinery [43]. Induced lysosomal damage can be accomplished by treating cells with the lysosomotropic compound L-Leucyl-L-leucine methyl ester (LLOMe) that disrupts the membrane of lysosomes [54]. We sought to determine whether treating infected cells

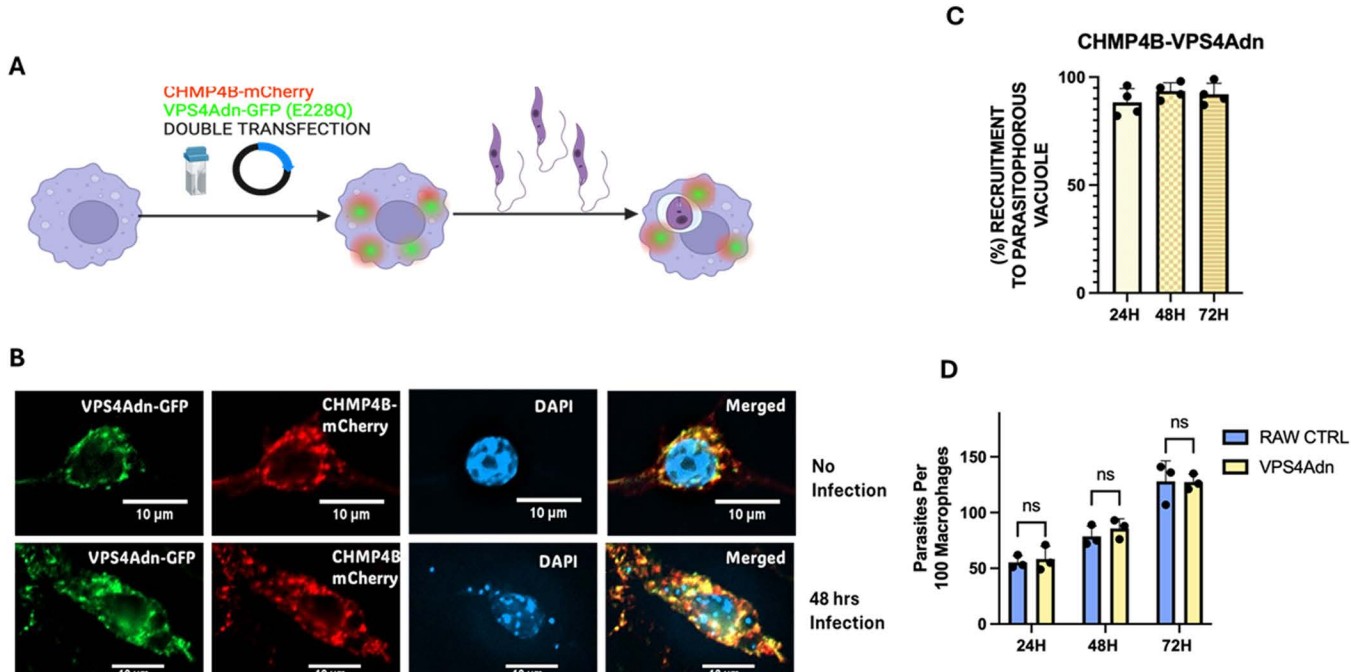

**Fig 2. Assessing the effect of VSP4-E228Q on the retention of ESCRT on LPVs.** (A). The scheme for assessing the effect of expressing the VPS4a dominant negative on ESCRT III distribution in transfected and infected cells. Macrophages were transfected with pEGFP-VSP4-E228Q and pLNCX2-mCherry-CHMP4B, plated on coverslips, and infected with *L. donovani*. Cells on coverslips were fixed at 24, 48, and 72 hours. B) Representative images of uninfected and infected cells are shown. (C) LPVs harboring parasites in VPS4-E228Q-transfected cells were scored for CHMP4B mCherry reactivity. (D) The course of infection in transfected cells was assessed at 24, 48, and 72 hours post-infection. Counts were obtained in duplicate on coverslips for at least three independent experiments. One-way ANOVA with multiple comparisons followed by *post hoc* Tukey's honest significant difference test to determine statistical significance was performed. *$p < 0.05$, **$p < 0.01$, ***$p < 0.001$, ****$p < 0.0001$. N. S, not significant. Graphs were generated in GraphPad Prism*8. Illustration was generated with Biorender.

with LLOMe would cause damage to LdLPVs, thereby inducing the recruitment of ESCRT components and activating the ESCRT machinery. The experimental scheme for LLOMe is illustrated in Fig 3A. CHMP4B-mCherry or CHMP2B-mCherry transfected cells plated on coverslips were infected with *L. donovani* for 48 hours. Thereafter, the cultures were pulsed with LLOMe (1mM or 5mM) or vehicle for 30 min. The cultures were washed and fixed at the indicated times. Almost all lysosomes within treated cells recruited CHMP4B-mCherry or CHMP2B-mCherry, which was monitored here as the positive indicator of LLOMe-induced damage. Representative images show that CHMP4B-mCherry is recruited to LdLPVs (Fig 3B). Approximately 95% of LdLPVs recruited CHMP4B-mCherry immediately after the pulse of 5mM LLOMe (Fig 3C, Fig 3D). The proportion of positive LdLPVs diminished rapidly when the drug was chased out. A significantly lower percentage of LdLPVs recruited CHMP2B-mCherry after 5mM LLOMe treatment than CHMP4B-mCherry (Fig 3E, Fig 3F). This observation suggested that ESCRT III components are differentially recruited to LdLPVs to repair induced damage. This pattern of differential recruitment of CHMP2b and CHMP4b contrasts with the earlier observations on the recruitment of these ESCRT III molecules to LdLPVs. Although there is the possibility that damage to LdLPVs may occur, which will necessitate repair, the LLOMe experiments suggest that recruitment for that purpose may be mechanistically distinct from

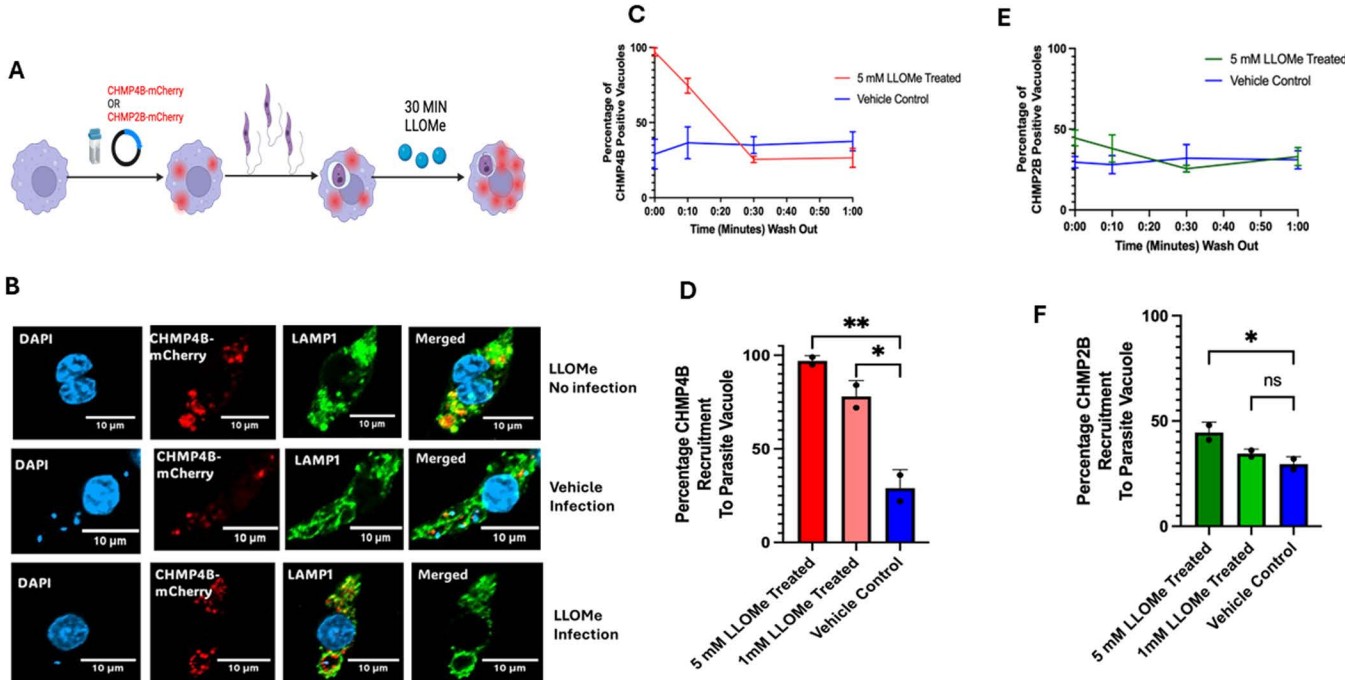

**Fig 3. LLOMe-induced damage of LdLPV suggests that damage-dependent ESCRT recruitment is mechanistically different from the 'normal' scheme of ESCRT recruitment to LdLPVs.** (A). The experimental scheme is shown. Cells were transfected with either pLNCX2-mCherry-CHMP2B or pLNCX2-mCherry-CHMP4B and plated on coverslips. They were then infected for 48 hrs, after which they were incubated with LLOMe (1mM or 5mM) or vehicle for 30 min. Cover slips were recovered when the drug was washed out (time 0) and then subsequently after 10-minute intervals. (B). Representative images of LLOMe-treated CHMP4B-transfected cells without infection are shown (red circles are due to CHMP4B in damaged lysosomes). Images from CHMP4B transfected and infected cells treated with either the vehicle or the drug are shown. LAMP1 labeling was used to delineate the contours of LPVs. (C) The number of LPVs that were positive for CHMP4B after the pulse and chase of LLOMe is shown. (D) A graph of the proportion of positive cells at the 0 time is shown. (E) The number of LPVs that were positive for CHMP2B after the pulse and chase of LLOMe is shown. (F) A graph of the proportion of CHMP2B-positive cells at the 0 time is shown. These graphs were compiled from two experiments, each with three coverslips for each point. Graphs were generated in GraphPad Prism*8 and statistical analysis was performed. A one-way ANOVA with multiple comparisons, followed by a post hoc Tukey's honest significant difference test, was performed to determine statistical significance. *p < 0.05, **p < 0.01, ***p < 0.001, ****p < 0.0001. N. S, not significant. Illustration was generated with Biorender.

the 'constitutive' recruitment of ESCRT molecules to LdLPVs, where CHMP2B and CHMP4B are recruited equivalently (Fig 1).

## The phosphoinositide PI(3,4)P2 on LdLPVs may be the target for recruitment of ESCRT molecules

Several studies have shown that phosphoinositides are differentially displayed on endomembranes [55]. PI4P, for example, is preferentially found associated with the Golgi. Recent studies showed that the phosphoinositide PI(3,4)P2 binds to ESCRT II and initiates the recruitment of the ESCRT machinery, which is indispensable for cytokinesis in the eye [47,56]. Zhang and colleagues [52] had shown that, like phagosomes that harbor inert particles, including Zymosan particles or latex beads, and display PI4P on their phagosomal membrane, *Leishmania* LPVs display PI(3,4)P2 in addition to PI4P [52]. In the studies here, we sought to affirm that LdLPVs display the relevant phosphoinositides, including PI(3,4)P2. Specifically, PI4P was detected with the Sidm probe (pMRXIP-GFP-P4M-SidMx2); PI(3.4)P2 was detected with the PI(3,4)P2 biosensor NES-eGFP-cPHx3, and PI(3,4,5)P3 was detected with the PIP3 biosensor (NES-EGFP-PH-ARNO2G-I303Ex2). Representative images of infected cells are shown (Fig 4). As an affirmation of these observations, infected cells recovered after 72 hours of infection with *L. donovani* were processed for the detection of PI(4)P, PI(3,4)P2, or PI(3,4,5)P3 using antibodies specific to each of these phosphoinositides (S2 Fig).

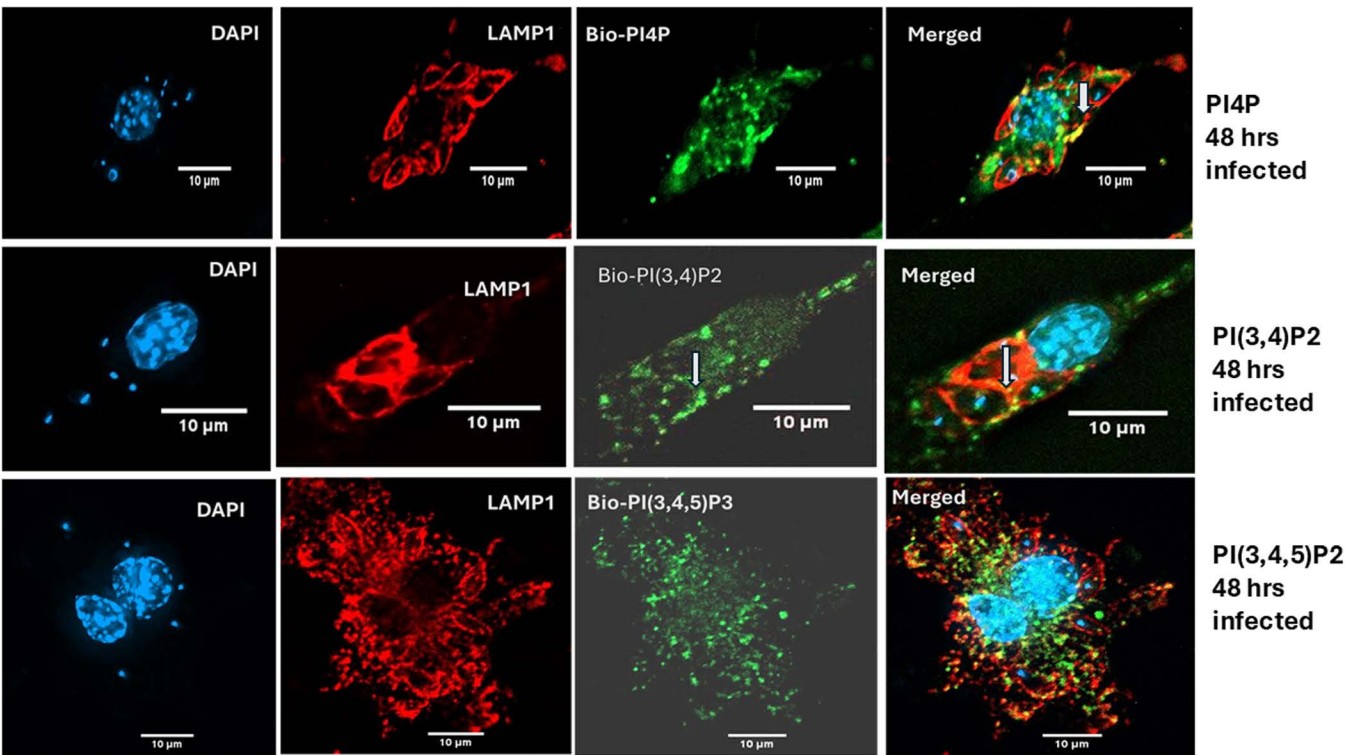

**Fig 4. PI(4)P and PI(3,4)P2 are displayed on LdLPVs.** RAW264.7 macrophages were transfected with pMRXIP-GFP-P4M-SidMx2 (PI4P biosensor), NES-eGFP-cPHx3 (PI(3,4)P2 biosensor), NES-EGFP-PH-ARNO2G-I303Ex2 (PI(3,4,5)P3 biosensor). Cells on coverslips were infected with *L. donovani*. Representative images of 48-hour-infected macrophages labeled for the detection of the biosensors and LAMP1 are shown. White arrows indicate the locations of the biosensor label. These images are representative of two experiments.

## Systematic knock-down of early components of the ESCRT machinery in RAW264.7 macrophages

What is the functional role of the ESCRT molecules that are recruited to LdLPVs? Several studies have described both canonical and non-canonical mechanisms for activating the ESCRT machinery. In cytokinesis, for example, studies have shown that ESCRT activation follows parallel schemes that are dependent on either the recruitment of ALIX or the recruitment of TSG101 to Cep55, which is followed by the recruitment of ESCRT III components in both schemes [26]. Studies on the budding of HIV from infected cells, in contrast, have shown greater dependence on the availability of ALIX as compared to TSG101 [57]. Informed by such observations, we proceeded to implement an experimental scheme in which we knocked down TSG101, VPS28, or ALIX and then monitored several parameters, including the recruitment of CHMP4B to LdLPVs, the division of LdLPVs, and the parasite burden of infected cultures. Knockdowns (KDs) were achieved by transfecting siRNAs (VPS28) or shRNA plasmids (TSG101 or ALIX) into macrophages, followed by selection in puromycin and evaluation of the resulting oligoclonal lines by Western blotting. Fig 5A shows the experimental scheme, which includes the assessment of cell viability of the selected lines. Representative Western blots in which TSG101 or ALIX was detected in the lysates from shCTRL, TSG101KD, or ALIXKD in RAW264.7 cells are shown (Fig 5B). Oligoclonal lines of ALIXKD and TSG101KD with greater than 60% knockdown were selected. We also selected cells transfected with shRNA control

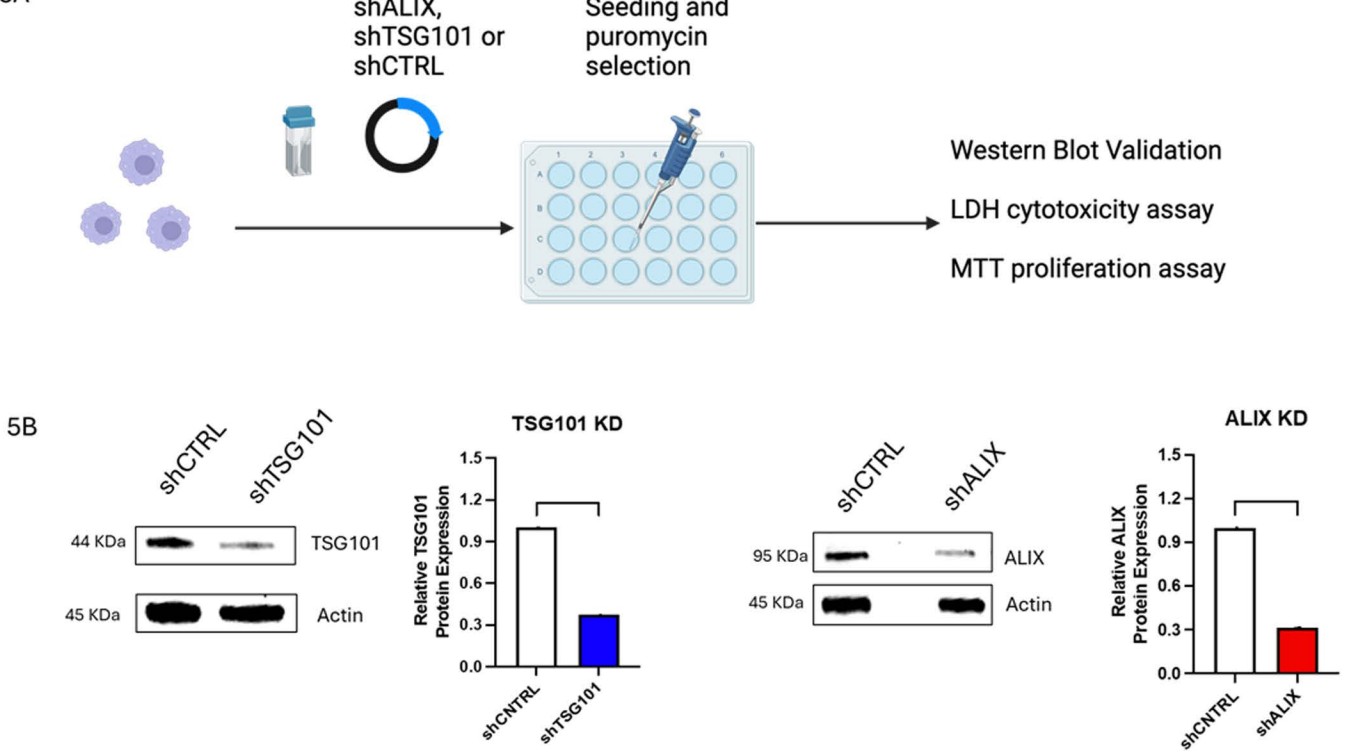

**Fig 5. Generation of cells expressing limiting levels of ESCRT components.** (A). The scheme for generating stable lines with knocked-down levels of TSG101 or ALIX is shown. Alix shRNA plasmid, or TSG101 shRNA plasmid, was transfected into macrophages and then plated. Twenty-four hrs after plating, the selection of cells by growth in puromycin commenced. Oligoclonal cell lines were identified, and the testing scheme for these cell lines is presented. The control line was transfected with a control short hairpin RNA (shRNA) plasmid. (B). Oligoclonal lines were tested for the expression of TSG101 or ALIX by Western blotting. Representative blots and densitometric analysis of blots to test for the expression of TSG101 or ALIX are shown. Plots were generated from densitometry of blots from at least 2 experiments. Graphs were generated in GraphPad Prism*8, where unpaired student *T* test statistical analysis was performed. *p < 0.05, **p < 0.01, ***p < 0.001, ****p < 0.0001. N. S, not significant. Illustration was generated with Biorender.

plasmid (shCTRL). Knocking down these molecules was not found to affect macrophage viability as measured in an LDH cytotoxicity assay and an MTT proliferation assay (S3A and S3B Fig).

## The knockdown of ALIX, unlike the knockdown of TSG101 or VPS28 had more noticeable effects on the infection

The KD cell lines and control lines were transfected with the CHMP4B-mCherry plasmid and then infected with Ld parasites. The experimental scheme is shown in Fig 6A. Representative images from the CHMP4B-mCherry transfections in RAW264.7 KD cells are shown in Fig 6B. Enumeration of the number of LdLPVs that recruited CHMP4B in shCTRL, TSG101KDs, and ALIXKDs is shown in Fig 6C. CHMP4B-mCherry was recruited to LdLPVs in shCTRL cells to similar levels as in 'wild type' infected cells. A similar proportion of LdLPVs recruited CHMP4B-mCherry in infected TSG101KDs. In contrast, CHMP4B-mCherry remained diffusely expressed in infected ALIXKDs, with limited recruitment to LdLPVs

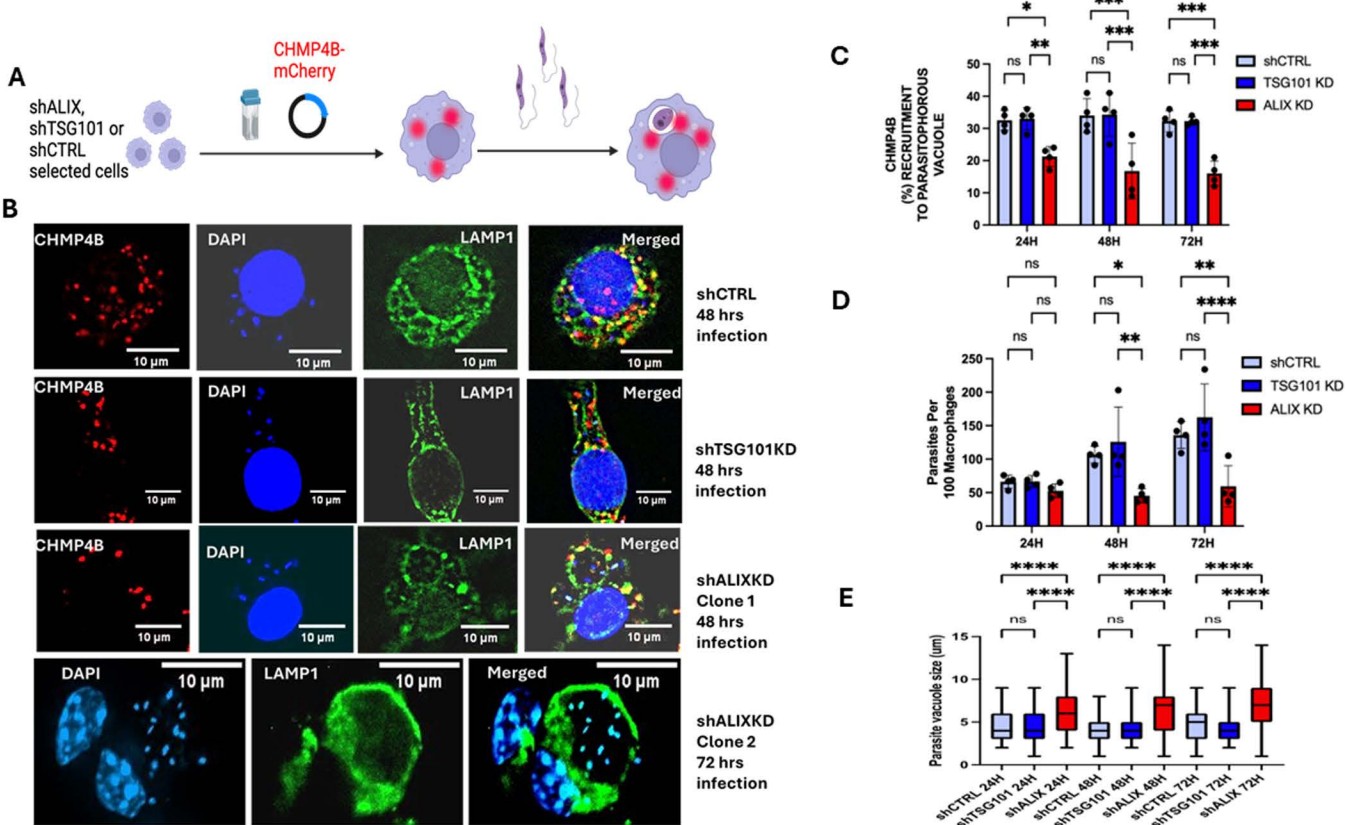

**Fig 6. Knockdown of TSG101 or ALIX in RAW264.7 macrophages reveals the critical role of ALIX.** (A). Selected oligo-clonal lines of RAW264.7 ALIXKD, TSG101KD, and shCTRL cells were transfected with the pLNCX2-mCherry-CHMP4B. They were then infected with *L. donovani* parasites. Coverslips were recovered after 24, 48, or 72h of infection and then processed to detect CHMP4B recruitment to LdLPVs. (B). Representative images from 48 hrs infected shCTRL, TSG101KD, or ALIXKD are shown. Contours of the LdLPV membrane were detected by LAMP1 labeling. The parasite and host cell nuclei were labeled with DAPI. ALIXKD clone 2 was not transfected with CHMP4B-mCherry. (C) CHMP4B recruitment to LdLPVs in KD cell lines was quantified as described in the Materials and Methods section. Colocalization of LAMP1 with CHMP4B was determined. At least 100 LdLPVs were measured per coverslip, per treatment, and time point. (D) The number of parasites per macrophage in infected KD cell lines was enumerated after 24, 48, or 72 hours of infection. At least 100 macrophages were counted per coverslip, treatment, and time point. (E) The LdLPV sizes in KD lines after 24, 48, and 72 hours of infection were measured and plotted. The diameter of LdLPVs was measured using the "Insert" scale tool of the BZ-X800 analyzer. At least 100 LdLPV sizes were measured per coverslip, time point, and treatment. Data was compiled and graphed using GraphPad Prism 8. One-way ANOVA with multiple comparisons followed by *post hoc* Tukey's honest significant difference test to determine statistical significance was performed. *$p < 0.05$, **$p < 0.01$, ***$p < 0.001$, ****$p < 0.0001$. N. S, not significant. Illustration was generated with Biorender.

(Fig 6B, Fig 6C). While approximately 33% of LdLPVs in shCTRL and in TSG101KDs were always CHMP4B-mCherry post-infection, less than 18% LdLPVs in ALIXKDs were positive for CHMP4B-mCherry. The number of LdLPVs in ALIXKD that were CHMP4b positive was significantly lower at 24, 48, and 72 hrs post-infection.

A commonly used indicator of *Leishmania* viability in infected cells is the demonstration that the number of parasites per macrophage increases over time, reflecting the replication of viable parasites. In contrast to infection in shCTRL and TSG101KDs, where there was the expected increase in the number of parasites/macrophages at 48 and 72 hrs post-infection, there were significantly fewer parasites/macrophages in ALIXKDs (Fig 6D). We then estimated the sizes of LdLPVs (Fig 6E). The average size of LdLPVs in shCTRL cells was 3.5 microns. In TSG101KDs, LdLPVs were also 3.5 microns. In contrast, the average size of LdLPVs in ALIXKD was approximately 6.8 microns, which was significantly larger than that of LdLPVs in shCTRL and TSG101KDs. As the plots show, in ALIXKDs, there is an increase in the proportion of much larger LdLPVs. Representative images of large LdLPVs in ALIXKDs that harbor numerous parasites are shown in Fig 6B. We elected to show Images from 2 ALIXKD lines generated from different transfections. Such larger LdLPVs harbored more than 4–10 parasites (S4 Fig). These results, including the images, demonstrate that with reduced availability of ALIX, the LdLPV division is defective, resulting in an approximately 70% reduction in the parasite burden in infected cultures. VPS28 was also knocked down by transfecting siRNA into RAW264.7 macrophages. Using this strategy, it was estimated that lysates prepared from transfected cultures contained significantly less VPS28 than mock-transfected control cells (S5 Fig). Enumeration of parasites per 100 macrophages showed a comparable number of parasites in cells with VPS28 knocked down as in mock control cells (S5B Fig). Estimation of the LdLPV sizes in infected cells after VPS28 knockdown showed that LdLPV sizes were comparable to the sizes of LdLPVs in mock control cells (S5C Fig). Taken together, these results, including the analysis of infections in VPS28 and TSG101 knockdowns, affirm the conclusion that the engagement of ALIX, but not the TSG101 or VPS28 (members of the ESCRT I complex), is required for the scission of LdLPVs. Scission of LdLPVs is essential for parasite replication within macrophages.

## ESCRT components are recruited to LdLPVs in infected THP-1 human cells

We elected to broaden the analysis of *L. donovani* replication in mammalian cells by performing experiments in THP-1 cells. The THP-1 cell line is a human peripheral blood-derived monocytic line that can be readily differentiated into macrophages. Differentiated THP1- cells do not proliferate, making them suitable for studies on replicating internalized *Leishmania* parasites. THP-1 cells are also amenable to genetic manipulation, including selection of new lines after the knockdown of genes of interest (GOI) and/or expression of tagged GOIs. To analyze the functions of the ESCRT machinery in THP-1 cells infected with *L. donovani* parasites, we tracked the distribution of representative ESCRT components in these infected cells. The recruitment of TSG101 (ESCRT complex I) and CHMP4B (ESCRT complex III) was monitored in infected cells by antibody detection in immunofluorescence assays. Representative images (Fig 7) are shown of TSG101 and CHMP4B-mCherry distribution in infected cells. Fig 7A shows a representative cell infected for 48 hrs labeled for detection of TSG101 (red label) or LAMP1 (green label). Individual parasites (blue label) in LAMP1-labeled LdLPVs are evident. We draw attention to two parasites in an LdLPV that are in the act of dividing (yellow arrow and inset). TSG101 is seen in the plane of scission that bridges the emerging daughter LdLPVs. Fig 7B shows the distribution of CHMP4B-mCherry in a 48-hour infected cell. An LdLPV that appears to be at an early stage of dividing is highlighted (yellow arrow and inset). Nucleation of CHMP4B is known to create membrane tension that precedes membrane scission [15,58]. In the figure, nucleated CHMP4B is at the center of an LdLPV between two parasites. Enumeration of LdLPVs that recruited TSG101 and CHMP4B-mCherry at the indicated times post-infection showed that approximately 40% of LdLPVs had recruited TSG101 or CHMP4B-mCherry (Fig 7C). This is consistent with observations in RAW264.7 macrophages reported above. Based in part on studies with VPS4A-E228Q, we concluded that ESCRT components are constitutively recruited to LdLPVs; however, levels of each ESCRT component may be below the detection threshold (Fig 1).

PLOS Pathogens

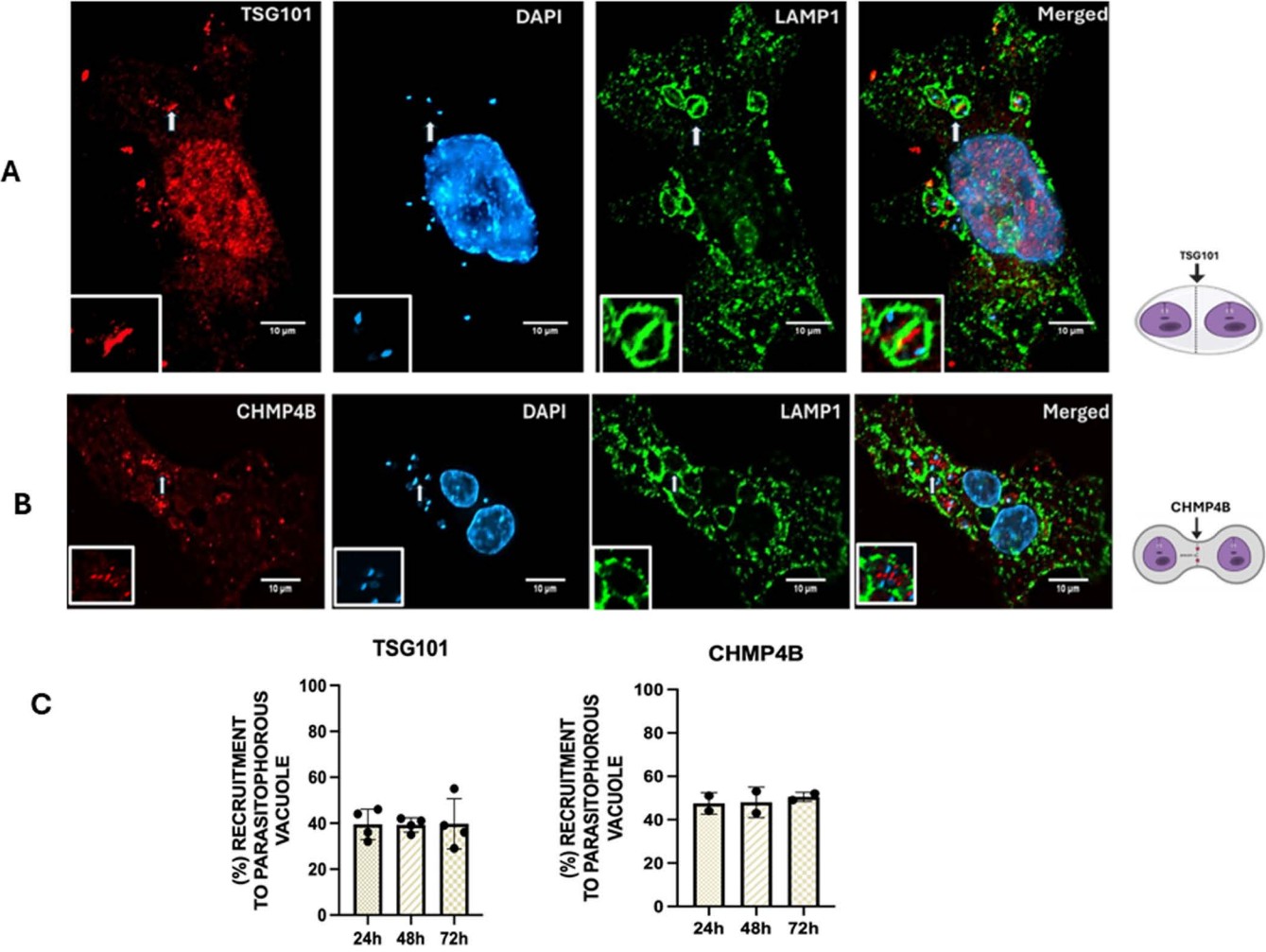

**Fig 7. The ESCRT machinery is recruited to LdLPVs in THP-1 cells.** After the differentiation of THP-1 monocytes into macrophages, they were infected with metacyclic promastigote forms of *L. donovani*. At 24, 48, and 72 hrs post-infection, cells on coverslips were fixed and processed by immunofluorescence labeling for the detection of representative components of the ESCRT I complex (TSG101) and the ESCRT III complex (CHMP4B). A) shows a representative cell labeled for the detection of TSG101 (red label), LAMP1 (green label), and parasite and host nuclei detected by DAPI labeling (blue label). An LdLPV that is dividing (yellow arrow and inset) is shown. TSG101 is in the plane of scission that bridges the emerging daughter LdLPVs. B) shows a representative cell labeled for the detection of CHMP4B. An LdLPV is highlighted (yellow arrow and inset) wherein nucleated CHMP4B is at the center of an LdLPV between two parasites. C) The plot shows the proportion of LdLPVs with the recruitment of TSG101 and CHMP4B. Counts were performed in duplicate on coverslips for at least three independent experiments. Illustration was generated with Biorender.

### *Leishmania* replication is impaired in THP-1 ALIXKD cells

We proceeded to generate THP-1 cells that stably expressed shRNA to either TSG101 or ALIX. Stable cell lines were selected and maintained by culturing them in puromycin, as described in the Materials and Methods section. Lysates from the TSG101KD or ALIXKD were analyzed in Western blots alongside lysates from cells transfected with the shCTRL plasmid (S6A Fig). The bar graphs plotted from a densitometric scan of the blots show that there was a greater than 70% reduction in expression of the TSG101 or the ALIXKD in KD cells.

CHMP4B and TSG101 recruitment to LdLPVs is significantly lower in the ALIXKD as compared to CHMP4B recruitment to LdLPVs in the shCTRL line or the TSG101KD cell line. A representative image of the distribution of CHMP4B-mCherry

in shCTRL, TSG101KD, and ALIX KD is shown in S6B Fig. A plot of the proportion of LdLPVs with CHMP4B-mCherry recruitment is shown. There are significantly fewer LdLPVs in ALIXKD-infected cells. The size of LdLPVs in shCTRL, TSG101KD, and ALIXKD cells was measured. S6C Fig shows that LdLPVs in ALIXKD were significantly larger than LdLPVs in shCTRL and TSG101KD cells. It is expected that in *Leishmania* infections with viable parasites, there is a continuous increase of parasites per macrophage over time. S6D Fig shows that parasites increase in the shCTRL and TSG101KD cell lines over 72 hours. In contrast, there is a limited, if any, increase in the number of parasites per cell in the ALIXKD line. The expectation is that in *Leishmania* infections with viable parasites, there is a continuous increase of parasites per macrophage over time.

To address the possibility that parasites in the ALIXKD were less viable, their expression of A2 was ascertained. A2 expression has been shown to commence several hours after metacyclic promastigote parasites are internalized within cells, and its expression is sustained throughout the intracellular amastigote stage [10,59]. Although there are more parasites in the shCTRL and TSG101KD lines as compared to infections in the ALIXKD line, the proportion of parasites expressing A2 was comparable (S7 Fig). We interpreted this result to mean that although the parasites were metabolically active, their capacity to undergo replication was defective. For an additional assessment of parasite viability in knocked-down lines, parasites were recovered from knocked-down cells, and their capacity to reinfect wild-type cells was ascertained. The data plotted in S8 Fig shows that infection with an equivalent number of parasites recovered from shCTRL, TSG101KD, or ALIXKD lines infected THP-1 macrophages comparably after 24, 48, or 72 hours of infection.

### BrdU incorporation into parasites revealed that Ld replication is impaired when LdLPV scission is inhibited

Amongst the reasons we elected to perform studies in the THP-1 cell line is that after differentiation into macrophages, THP-1 cells cease to proliferate. It is, therefore, feasible to perform analyses that monitor parasite replication with a reagent such as BrdU without contending with interfering signals from the nucleus of host cells that are actively multiplying, as is the situation with RAW264.7 mouse macrophages. Following the infection of macrophages by the infective metacyclic promastigote form of *Leishmania*, the parasites undergo amastigogenesis, characterized by changes in gene expression and metabolism that culminate in the initiation of replication approximately 24 hours later [9]. In *L. donovani* infections, daughter parasites segregate into separate LdLPVs. The studies above have shown that the host cell's ESCRT machinery appears to play a crucial role in the scission of LdLPVs, which facilitates the division of the daughter cells. In ALIXKD cultures, there are significantly fewer parasites, suggesting that the parasites either died or failed to replicate in the absence of ALIX. To specifically monitor parasite replication, the incorporation of BrdU into intracellular parasites was assessed. The expectations for these studies are shown in Fig 8A. After a pulse of BrdU for up to 24 hrs post-infection, healthy parasites would be expected to incorporate BrdU. As the parasites replicate, they should lose BrdU upon subsequent division. In these studies, BrdU was added to the culture medium after 6 hours of infection. After 24 hours of infection, the medium was replaced, and infections continued until 72 hrs post-infection. The representative images (Fig 8B) show that most parasites incorporate BrdU after 24 hrs. By 72 hrs post-infection, few parasites in the shCTRL or TSG101KD cells retained BrdU. In contrast, parasites in ALIXKD cells remain BrdU positive. Enumeration of the proportions of BrdU-labeled parasites confirmed that by 48 hrs post-infection (Fig 8C), parasites in shCTRL and TSG101KD cells lose BrdU. In contrast, there are significant numbers of parasites that retain BrdU in the ALIXKD. LD replication is impaired in cells with reduced ALIX levels that have a defect in LDLPV scission and division.

### Discussion

*Leishmania donovani* (Ld) is the causative agent of visceral leishmaniasis, which is fatal if not treated. In mammalian cells, Ld are housed individually within membrane-enclosed compartments that enigmatically divide to accommodate each daughter parasite after parasite replication. In this study, we showed for the first time that components of the ESCRT machinery are recruited constitutively to LdLPVs. There, the ESCRT machinery catalyzes the scission of LdLPVs, which

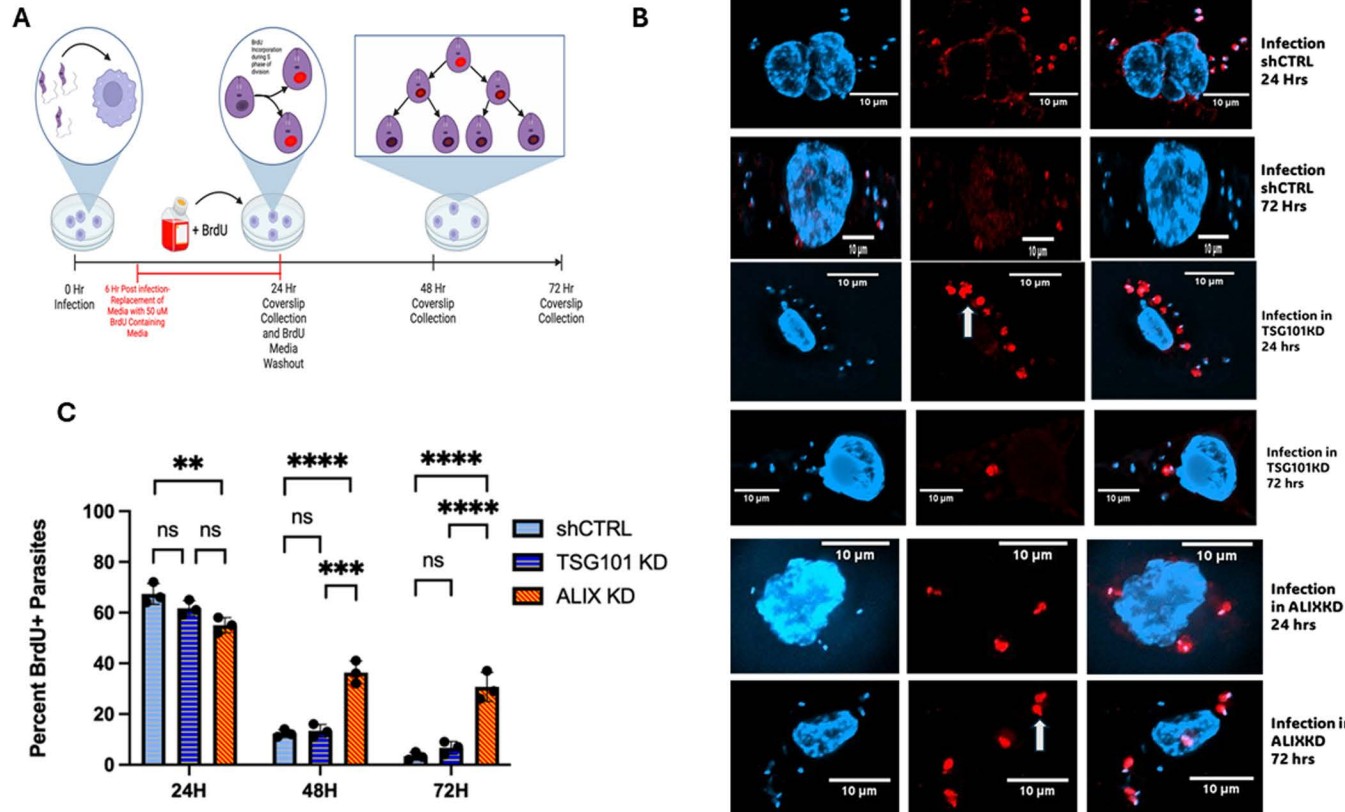

**Fig 8. BrdU incorporation into intracellular parasites shows that parasite replication is impaired in ALIXKD.** (A). The experimental scheme is shown. After 6 hrs of infection, BrdU is added to the culture medium. After 24 hrs of infection, the culture medium was washed out and replaced with BrdU-free medium. Parasites that initiate replication are expected to incorporate BrdU, which is lost upon successive rounds of replication. (B) Representative images of microscopic evaluation of BrdU incorporation (red label) at 24 and 72 hrs in shCTRL, TSG101KD, and ALIXKD are shown. White arrows in the images point to representative examples of labeled parasites. C) The percentage of parasites that were labeled positively for BrdU at the indicated times was plotted. At least 100 parasites were counted per coverslip, treatment, and time point. A one-way ANOVA was performed, and statistical significance was determined by *post hoc* Tukey's honest significant difference test. Data was compiled and graphed using GraphPad Prism 8. *$p < 0.05$, **$p < 0.01$, ***$p < 0.001$, ****$p < 0.0001$. N.S, not significant. Illustration was generated with Biorender.

is essential for completing Ld replication. Our studies revealed a chain of events that may commence with the display of the phosphoinositide, PI(3,4)P2, on LdLPVs, followed by the recruitment of ALIX to LdLPVs. Impairment of ALIX recruitment by limiting its availability results in bloated LdLPVs that house 3 or more parasites due to impaired LdLPV scission. Interestingly, although the initiation of parasite replication is not dependent on the dynamics of the ESCRT machinery, continuous replication of Ld in infected macrophages is inhibited when activation of the ESCRT machinery is defective. Together, these results show that ALIX plays a central role in recruiting the ESCRT machinery to LdLPVs where it catalyzes the scission of the LdLPV pseudo-organelle. That a non-canonical activation of the ESCRT machinery plays a role in the division of LdLPVs is a new function for the ESCRT machinery.

To gain more insight into the recruitment of ESCRT components to LdLPVs, we incubated infected cells in LLOMe. This compound is a potent lysosomotropic agent that disrupts the lysosomal membrane. Several studies have investigated the consequences of exposing cells to this compound [60,61]. We observed that a 30-minute incubation with LLOMe caused rapid damage to most lysosomes, as indicated by the recruitment of CHMP4B-mCherry and CHMP2B-mCherry to these compartments. Although most LdLPVs in treated cells became CHMP4B-mCherry positive in response to LLOMe

treatment, many fewer LdLPVs were CHMP2B-mCherry positive. This difference in the recruitment of these two ESCRT III components contrasts with what we observed in studies on the recruitment of CHMP2B and CHMP4B to LdLPVs in infected cells (Fig 1). We interpreted these results to mean that ESCRT III recruitment to LdLPVs in infected cells is mechanistically different from the recruitment of ESCRT III molecules to damaged lysosomes or lysosome-like membranes. Additional discussions and insights into lysosome damage and repair mechanisms, gleaned from studies with LLOMe, are available [60,62].

Studies that assessed the incorporation of BrdU into intracellular parasites were illuminating. It is known that after internalization of the metacyclic promastigote form of *Leishmania spp.* they gradually transform into the aflagellate amastigote form [9]. Incorporation of BrdU after approximately 24 hrs post-infection is an accepted indicator of parasite replication [9]. We demonstrated that BrdU is incorporated to comparable levels in shCTRL cells, TSG101KD cells, and ALIXKD cells at 24 hours post-infection. While BrdU becomes increasingly less detectable in parasites within shCTRL and TSG101KD macrophages as the infection progresses, Ld in ALIXKD cells retains BrdU at 48 and 72 hrs post-infection. These studies were interpreted to mean that in ALIXKD cells, parasite replication is impaired. Notably, Ld parasites in the ALIXKD continuously expressed A2, a well-characterized antigen expressed by viable intracellular parasites, which argues against the likelihood that the failure to proliferate was due to parasites dying in infected cells.

There is still uncertainty on the range of conditions that trigger the recruitment of ESCRT components to membranes. Membrane damage caused by rupture of endomembranes, including the insertion of secretion apparatus into membranes by intracellular pathogens, is a reliable trigger for the recruitment of ESCRT components [38,42]. Current understanding of LdLPV biology has not revealed the existence of a secretion apparatus or a process that may cause damage to the LdLPV and, by so doing, activate the recruitment of ESCRT components. Instead, we previously showed that phosphoinositides, specifically PI(3,4)P2 are displayed on LPVs [52]. Informed by studies that have identified PI(3,4)P2 and other phosphoinositides as membrane anchors for the recruitment of ESCRT components [47,48] we affirmed that LdLPVs do indeed display PI(3,4)P2. Future studies will attempt to identify the specific ESCRT molecule(s), including ALIX, that are initially recruited to the LdLPV and the conditions that lead to non-canonical activation of the ESCRT machinery. This parasite likely generates additional signals that instruct the machinery to execute the fission and scission of LPVs, a phenomenon that does not occur with other pathogen-containing compartments.

The membrane anchor for ESCRT components is a crucial issue that has been considered in several pathogen-host interactions. In the budding of HIV, the Gag protein of the virus was shown to be the primary viral molecule that interacts with ESCRT [63,64]. Following such studies, small molecules that selectively block the HIV Gag – ALIX interaction have been proposed as potential viable inhibitors for the spread of HIV [65]. In *Toxoplasma* infections, proximity labeling studies revealed interactions between ALIX and so-called GRA proteins discharged from the parasites' Rhoptry organelle that are displayed on the parasite vacuole [44]. Other studies had implicated the interactions of TSG101 in interactions with other GRA proteins that were necessary for nutrient acquisition into the parasitophorous vacuole [45]. Interactions between ESCRT components and parasitophorous vacuoles may satisfy different functions.

Based on our observations, we suggest that ESCRT components are recruited constitutively to LdLPVs and are anchored to phosphoinositides displayed on the LdLPV membrane. The parasite may play an essential role in the synthesis of phosphoinositide displayed on the LdLPV membrane. We determined that ALIX plays a critical role in the initial anchoring of the ESCRT machinery to LdLPVs. However, we do not exclude the possibility that other ESCRT components are recruited in conjunction with ALIX. The recruitment of CHMP4B and CHMP2B follows the recruitment of ALIX. Subsequent activation of the ESCRT machinery, characterized by the aggregation and/or nucleation of ESCRT III, occurs during parasite division. We observed CHMP4B, most prominently, at the bridge between two LdLPVs, presumably at the point of scission (Fig 1, Fig 7). In the graphical abstract (Fig 9), we highlight that when ALIX levels are limiting, LdLPVs are increasingly enlarged and can harbor greater than 3 parasites. There is a defect in the assembly of the ESCRT machinery, resulting in the inability to execute the scission of the LdLPV. Some outstanding questions include: What could be the

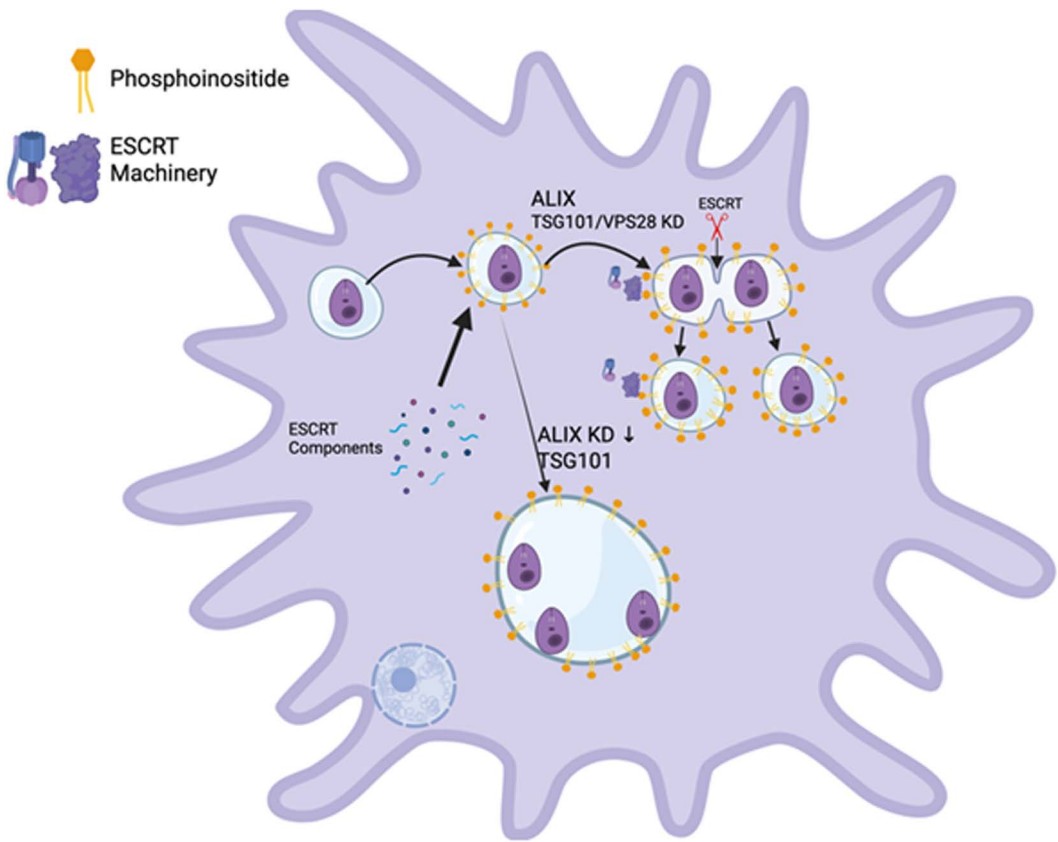

**Fig 9. Graphical summary: LdLPVs display phosphoinositides that are the membrane anchors for components of the ESCRT machinery.**
ALIX is the critical component that is recruited. Members of the ESCRT I complex are recruited, but their recruitment is dispensable. Components of the ESCRT III complex and VPS4 are recruited subsequently to complete the activation of the ESCRT machinery. The scission of LdLPVs completes the replication of parasites and LdLPVs. When ALIX expression is limited, LdLPV fission is defective, and so too is parasite replication. This results in a reduction in the parasite burden. Illustration was generated with Biorender.

signal(s) for LdLPV scission? Does LdLPV scission share mechanistic details with cytokinesis? Alternatively, LdLPV scission could share signal(s) with the initiation of viral budding, where the interactions of ALIX and the Gag protein are critical for the progress of that process. Future studies will likely address these questions.

## Materials and methods

### Parasite culture

*L. donovani* (MHOM/S.D./62/1S-CL2$_D$) was obtained from Dr. Nakhasi's lab (FDA) and cultivated in M199 media (Sigma) containing 20% FBS, 0.1 mM Adenosine, 0.1 mg/mL folic acid, 2 mM glutamine, 25 mM HEPES, 100 units/mL penicillin/-100 µg/mL streptomycin 15,140,122 (Gibco), 1X BME vitamins (Sigma), and 1 mg/mL sodium bicarbonate with pH 6.8 at 26°C.

### Mammalian cell culture

RAW264.7 macrophages were obtained from ATCC (TIB-71) and maintained in DMEM (Corning) supplemented with 10% FBS and 100 units/mL penicillin/-100 µg/mL streptomycin 15,140,122 (Gibco). Cells were kept in complete DMEM at

37°C in a humidified atmosphere incubator containing 5% $CO_2$. THP-1 suspension monocytes were obtained from ATCC (TIB-202) and maintained in RPMI 1,640 (Corning) supplemented with 10% FBS and 100 units/mL penicillin/-100 µg/mL streptomycin 15,140,122 (Gibco) at 37°C in a humidified atmosphere incubator containing 5% $CO_2$.

## Macrophage infections

RAW264.7 macrophages were plated in culture dish containing sterile glass coverslips and incubated overnight at 37°C with 5% $CO_2$. THP-1 suspension monocytes were differentiated into macrophages by culturing them in complete media culture dishes containing sterile glass coverslips supplemented with 10ng/mL of phorbol 12-myristate 13-acetate (PMA) J63916.MCR (ThermoFisher Scientific) for 24 hours at 37°C with 5% $CO_2$. PMA was washed out with complete media before Ld infection. For infection, metacyclic promastigotes were selected from late stationary stage parasites using pea-nut agglutination (PNA) according to an established protocol [49]. Briefly, 4-day old cultures of *L. donovani* parasites were washed twice and resuspended in incomplete DMEM at a concentration of $2\times10^8$ parasites/mL. PNA was then added to the parasites at a final concentration of 50 µg/mL and incubated at room temperature for 15 mins. The parasites were then centrifuged at 200x*g* for 5 mins to pellet agglutinated parasites. The supernatant was then collected and the PNA- metacyclic parasites were washed twice, resuspended in complete DMEM, and counted for infection. Parasites were then added to macrophage dishes at a ratio of 5:1 MOI for THP-1 macrophages or 10:1 MOI for RAW264.7 macrophages (MOI = parasites to macrophages) then incubated at 37°C with 5% $CO_2$. Infections were stopped at 24, 48, or 72 hours post-infection by placing coverslips in 4% paraformaldehyde for 20 min for fixation at room temperature. After fixation, cells were ready for the immunofluorescence assay. For experiments to evaluate the recruitment of proteins of interest to generic phagosomes, latex beads L0280 (SIGMA-ALDRICH) were incubated with macrophages at an MOI equal to parasite infections

## Generation of transfectants and knockdowns in RAW264.7 cells and THP-1 cells

To develop cells that express fluorophore-tagged variants of ESCRT molecules, we acquired the following plasmids: pLNCX2-mCherry-CHMP2B was a gift from Sanford Simon (Addgene plasmid # 115,331), pLNCX2-mCherry-CHMP4B was a gift from Sanford Simon (Addgene plasmid # 116,923), pLNCX2-mCherry-VPS4A was a gift from Sanford Simon (Addgene plasmid # 115,334), pEGFP-VSP4-E228Q was a gift from Wesley Sundquist (Addgene plasmid # 80,351), pLNCX2-mEGFP-TSG10 was a gift from Sanford Simon (Addgene plasmid # 116,925). To develop cells expressing biosensors to detect PI4P, PI(3,4) P2, and PI(3,4,5)P3, we acquired the following plasmids: pMRXIP-GFP-P4M-SidMx2 PI4P sensor was a gift from Noboru Mizushima (Addgene plasmid # 221,747), NES-eGFP-cPHx3 PI(3,4)P2 sensor was a gift from Gerry Hammond (Addgene plasmid # 116,855), NES-EGFP-PH-ARNO2G-I303Ex2 PI(3,4,5)P3 sensor was a gift from Gerry Hammond (Addgene plasmid # 116,868).To approximately $1\times10^7$ RAW264.7 cells, 10ug plasmids were introduced by nucleofection using reagents from Mirus Bio (MIR 50,112) (AMAXA Nucleofector II). For the most part, cells were used in experiments after transient transfections. Twenty-four hours after nucleofection, cells were infected with *L. donovani* parasites or incubated with latex beads as described above. To develop stable knockdown cell lines, the following plasmids were obtained: TSG101 shRNA Plasmid sc-36753-SH (Santa Cruz), Alix shRNA Plasmid sc-60150-SH (Santa Cruz), and Control shRNA Plasmid sc-108060-SH (Santa Cruz). After nucleofection (AMAXA Nucleofector II), cells were seeded in 24 well plates for selection of oligoclonal lines. After 24 hours, the culture medium was replaced with medium containing 6ug/mL of puromycin (it was predetermined that the RAW264.7 wild type cells were killed by 4ug/mL puromycin). Colonies were selected from the 24-well plates and expanded into 6-well plates. Oligoclonal lines in 6-well plates were expanded for downstream application and experimentation. For VPS28 knockdown RAW264.7 experimentation, we generated a transient knockdown using VPS28 siRNA sc-41101 (Santa Cruz) and used a mock RAW264.7 transfection without plasmid as a control. For THP-1 suspension monocytes, cells were treated as described above for the generation of targeted protein knockdown cell lines, with a puromycin concentration of 1ug/mL determined to be optimal

for selection. THP-1 monocytes were cultured in complete media supplemented with selection antibiotic for 4 weeks until reaching confluency. Complete media with the selection antibiotic was periodically changed. Knock down of specific proteins was validated by Western Blotting.

## LDH cytotoxicity and MTT proliferation assays on protein-knockdown cells

To test if cells that were ALIX or TSG101 knocked down in both RAW264.7 and THP1 macrophages were viable, we performed an LDH (Lactate dehydrogenase) cytotoxicity assay with the CK12 (DOJINDO) cytotoxicity kit. 5,000 cells were seeded in 96 well plates in 4 wells per cell line, and 96 well plates were for each time point. Then, the assessment of LDH release was followed per kit instructions, and cytotoxicity was determined using LDH release from cells that were transfected with the control plasmid as a reference. For the MTT proliferation assay, 2000 cells were seeded in 96-well plates in 4 wells per cell line, and 96-well plates were separated for each time point. Proliferation was assessed using the MTT cell proliferation kit AR1156 (BOSTER), and the kit protocol was followed for each collected time point.

## Immunofluorescence assay (IFA) and fluorescence microscopy

Glass coverslips with infected macrophages were fixed in 4% paraformaldehyde (PFA) at the indicated times post-infection. PFA was washed out after 20 min of incubation and then washed with 1xPBS twice. Coverslips were left in 1XPBS until all timepoints of infection were collected for downstream application. Coverslips were then processed in immunofluorescence assays (IFAs) to visualize the distribution of LAMP1 to delineate parasite vacuoles, PI(3,4)P2 for phosphoinositides. Anti-LAMP1 1D4B (DSBH) was used for RAW264.7 mouse-derived macrophages, and anti-LAMP1 G1/139/5 (DSBH) or W18263B (BioLegend) was used for THP-1 human-derived macrophages. Anti PI4P Z-P004 (Echelon Biosciences), anti PI(3,4)P2 Z-P034 (Echelon Biosciences), anti PI(3,4,5)P3 Z-P345 (Echelon Biosciences), anti-TSG101 ZE4325959 (Invitrogen), anti-CHMP4B SAB2105901 (SIGMA), anti-*Leishmania* A2 AB150344 (abcam) and anti-BrdU 14-5071-82 (Invitrogen) primary antibodies were diluted at a 1:50 concentration for anti-LAMP1 antibodies and a 1:200 concentration for every other antibody. Chicken anti-rat IgG AF594 conjugated A21471 (Invitrogen), goat anti-rat IgG AF488 conjugated A11006 (Invitrogen), goat anti-rabbit IgG AF594 A32740 (Invitrogen), rabbit anti-mouse IgG AF594 conjugated A11062 (Invitrogen) and chicken anti-mouse IgG FITC conjugated NB120−6,810 (NOVUS) secondary antibodies were diluted in 1xPBS at a 1:200 concentration. Cells were permeabilized in 0.5% Triton-X-100 in 1XPBS for 15 min and blocked in 5% bovine serum albumin (BSA). In the case of phosphoinositide lipid staining, 0.5% saponin was used for permeabilization, as recommended by the manufacturer, for 15 min. Incubation in primary antibodies diluted in 1XPBS with 1% BSA was performed for 1 hour at room temperature. After washing in 1XPBS, coverslips were incubated in AlexaFluor/FITC conjugated secondary antibodies for 30 min at room temperature. Coverslips were washed and then mounted on glass slides using the ProQ diamond mounting agent supplemented with 4′,6-diamidino-phenylindole (DAPI) stain. Coverslips were visualized and captured using BZ-X810 (Keyence, Osaka, Japan) All-in-One Fluorescence Microscope at 100x oil immersion objective magnification or Zeiss Axio Observer Z1/7; 63×water immersion objective magnification. Images were captured with an Axiocam 503 controlled by Zen Acquisition software. Scored LdLPVs were delimited by Lamp1 reactivity that contained at least one parasite nucleus. The percentage of infected macrophages and the average number of parasites were determined by counting at least 100 macrophages per coverslip for LdLPVs or latex bead phagosomes. Counts were done in duplicate coverslips for at least 3 independent experiments.

## BrdU labeling

After 6 hrs of infection, cultures were incubated in culture medium supplemented with 50uM 5-Bromo-2′-Deoxyuridine (BrdU) B23151 (Invitrogen). After 24 hrs infection, the BrdU containing media was replaced with complete media without BrdU. At indicated times, coverslips were recovered and fixed in PFA. Thereafter, they were washed in 1XPBS, then incubated for 10 min in 50mM ammonium chloride to quench the remaining PFA. They were then incubated with 2 N HCl

for 40 min to denature the nucleic acids to expose the incorporated BrdU. Coverslips were then washed with 1XPBS for 20 min to remove HCl after which we proceeded to typical cell permeabilization.

## Extraction of amastigotes for parasite viability assessment

Macrophages were infected with Ld as described above. Recovery of parasites for viability analysis was performed according to published protocols [66]. Seventy-two hours after infection, amastigotes were extracted by removing the supernatant from infected plates and adding serum free RPMI media with 0.05% SDS and shaken for 30 seconds. After lysis of macrophages, amastigotes were quickly recovered by centrifuging the supernatant at 2,500 RPM for 10 min. Amastigotes extracted from KD and control macrophages were counted, and an equal number of amastigotes was used to infect wild-type THP-1 macrophages. Coverslips were recovered at 24, 48, and 72 hours post-infection to assess the viability of parasites.

## Image analysis

Images were analyzed using the BZ-X800 analyzer for pictures taken in the BZ-X810 fluorescent microscope, or Zen Blue 2.6 for images taken in the Zeiss Axio observer Z1/7. Z-stack projections were used for specific image analysis of colocalizing proteins. Using Fiji/ImageJ, a mask was created around parasite vacuoles or latex bead phagosomes, determined to be regions of interest (ROI). The DAPI signal of the parasites, as well as the blue fluorescence of the latex beads was used to determine the ROI. The purpose of creating an ROI mask around the phagosome of a parasite or bead is to exclude cytosolic or other compartmental protein signals from the colocalization analysis. Then, the BIOP JAcOP colocalization of signals plug-in was used to determine if the signals of proteins of interest colocalized in the ROI previously selected using the masking option. LAMP1 reactivity was used to delineate the vacuole of the parasite or the latex bead phagosome; colocalization of LAMP1 with ESCRT proteins of interest was performed. If Pearson's correlation coefficient, as yielded by JACoP, met a specific threshold of 0.4 or higher for the masked region, it was considered positive. At least 100 parasite vacuoles or latex bead phagosomes were counted per coverslip, per treatment, and time point. This analysis was applied to each experimental scheme.

## LLOMe pulse/chase assay

To determine the impact of L-Leucyl-L-Leucine methyl ester (LLOMe) hydrobromide HY-129905A (MedChemExpress) on parasite vacuoles, we prepared a stock solution of 100 mM using dimethyl sulfoxide (DMSO) as the solvent. LLOMe was diluted with complete DMEM at 5mM and 1mM concentrations and added to infected macrophages 48 hours post-infection. Infected cells were stimulated for 30 min with each LLOMe concentration, and an equal amount of DMSO was added to complete the DMEM for the vehicle control, which was added simultaneously in separate vessels. After 30-minute exposure, LLOMe was washed out twice with complete DMEM, and coverslips were collected post LLOMe stimuli at 0 min, 10 min, 30 min and 60 min after treatment.

## Parasite vacuole measurement, parasites per vacuole and parasites per macrophage counts

Z-Stack images of infected cells on coverslips were acquired as described above. Z-Stack heights were selected to cover the entire host cell nucleus and any PVs being measured to ensure that the widest point of the host cell nucleus and PV was captured. Using the "Insert" scale tool of the BZ-X800 or ZEN ZEISS analyzer, the diameter of parasite vacuoles was measured. At least 100 parasite vacuole sizes were measured per coverslip, per treatment, and time point. For parasite counts per macrophage, at least 100 macrophages were counted per coverslip, per treatment, and time point. Parasites per vacuole were counted using DAPI parasite nuclei, and Lamp1 reactivity for parasite vacuole reference.

## Western blotting

Cells were lysed in RIPA buffer #89,900 (Thermo Scientific) and protein concentration was measured by BCA assay #23,225 (Thermo Scientific). Aliquots with equal protein (ug) amounts were suspended in SDS PAGE loading buffer and

run at 140V for 1hr. Protein was transferred onto a nitrocellulose membrane 10,600,006 (GE Healthcare) or polyvinylidene difluoride membrane IPVH15150 (Immobilon) with 20V for 10hr in 4°C. Primary antibodies included anti-Alix sc-53540 (Santa Cruz), anti-TSG101 PA5–81,094 (Invitrogen), VPS28 sc-166537 (Santa Cruz), and anti-Beta Actin sc-47778 (Santa Cruz). Secondary HRP conjugated antibodies included Goat anti-mouse IgG 31,430 (Invitrogen) and Chicken anti-Rabbit IgG A15993 (Invitrogen). Antibody dilutions were prepared in 1X TBST (Tris buffered saline Tween 20) with 5% BSA for primary antibodies, and 5% non-fat milk for secondary antibodies. Primary antibodies were diluted at a 1:1000 concentration, and secondary antibodies at a 1:2000 concentration. Membranes were blocked with 5% non-fat milk for 1hr then incubated with primary antibody overnight at 4°C in constant rocking. After washing with TBST 3 times for 10min each, secondary antibody incubation was performed for 1 hour. Proteins of interest were then visualized using chemiluminescence on the Invitrogen iBright imaging system (Thermo Fisher Scientific). Protein quantification was performed using FIJI/imageJ's densitometric analysis tool.

## Statistical analysis of data

After data acquisition and data collected by colocalization of proteins of interest, parasite vacuole sizes, parasite infectivity, parasites per vacuole and parasites per macrophage were graphed using GraphPad Prism 8. Data was first imported, then a one-way ANOVA or unpaired Student *T* Test was performed, and statistical significance was determined by *post hoc* Tukey's honest significant difference test in the case of ANOVA. Similarly, densitometry data for western blot quantification of protein levels was acquired from FIJI/ImageJ as described above, then GraphPad Prism 8 was used to generate bar graphs and perform unpaired student *T* Test. Significance was determined by a p-values of $*p < 0.05$, $**p < 0.01$, $***p < 0.001$, $****p < 0.0001$

## Supporting information

**S1 Fig. ESCRT components are not recruited to phagosomes that harbor inert particles.** Recruitment of ESCRT components to generic phagosomes was determined by incubating transfected cells with latex beads. A representative image of a cell transfected with CHMP2B and a phagocytosed latex beads is shown. The proportion of phagosomes that displayed CHMP2B or CHMP4B is shown in the associated graphs. LAMP1 reactivity was used to delineate the vacuole of the LPV or latex bead phagosome. White arrow points to a latex bead.
(TIF)

**S2 Fig. PI(4)P and PI(3,4)P2 are displayed on LdLPVs.** THP-1 cells on coverslips were infected with *L. donovani*. After 48 hours coverslips were fixed and processed for immunofluorescence detection of phosphoinositides with antibodies to each phosphoinositide. Representative images of THP-1 cells infected for 48 hours are shown. These images are representative of two experiments.
(TIF)

**S3 Fig. Neither TSG101KD nor ALIXKD is cytotoxic to RAW264.7 cells.** (A). Cytotoxicity was determined by monitoring LDH release from oligoclonal KD cells and from shCTRL. For the MTT proliferation assay, cells were seeded in 96-well plates, with 4 wells per cell line, and separate 96-well plates were used for each time point. (B) Proliferation was assessed using the MTT cell proliferation kit AR1156 (BOSTER). The results are from 2 experiments.
(TIF)

**S4 Fig. Increased number of parasites per vacuole in ALIXKD cells.** Infected cells on coverslips were fixed after 24, 48, and 72 hours of infection. Immunofluorescence labeling of LAMP-1 permitted detection of the contours of LPVs. Parasite nuclei were detected with DAPI. The number of parasites per LPV was counted at 24, 48, and 72 hours after infection. At least 100 LdLPVs were measured from each coverslip. Data was compiled from at least 3 experiments and graphed using GraphPad Prism 8.
(TIF)

**S5 Fig. Knockdown of VPS28 does not affect *L. donovani* infection of RAW264.7 macrophages.** RAW264.7 cells were transfected with siRNA to VPS28 and plated in dishes with or without coverslips. Control cells were mock-transfected. Lysates were prepared from dishes without coverslips. Lysates were analyzed by Western blotting for VPS28 expression (A). Plot from the densitometric analysis is shown. B) Cover slips were infected with *L. donovani* parasites for 24, 48 or 72hs. Immunofluorescence labeling was performed to evaluate the infection. The number of parasites per macrophage in infected VPS28 KD cultures, as compared to infections in mock controls, was enumerated and plotted. Data was compiled and graphed using GraphPad Prism 8. C) LAMP1 labeling was performed to confirm LdLPVs. The LdLPV sizes in KD lines after 24, 48 and 72 hrs infection were measured and plotted. At least 100 LdLPV sizes were measured per coverslip. A one-way ANOVA was performed, and statistical significance was determined by *post hoc* Tukey's honest significant difference test. For western blotting validation, an unpaired student *T* test was performed for statistical analysis. Data were compiled and graphed using GraphPad Prism 8. *$p < 0.05$, **$p < 0.01$, ***$p < 0.001$, ****$p < 0.0001$. N.S, not significant.
(TIF)

**S6 Fig. Knockdown of ALIX or TSG101 in THP-1 cells and evaluation of *L. donovani* infections.** The TSG101 shRNA plasmid, Alix shRNA plasmid, and Control shRNA plasmid were transfected into THP-1 monocytes. Stably transfected cells were selected by growth in media supplemented with puromycin. Lysates from stable lines were analyzed in Western blots. A) Representative blots were probed for TSG101 or ALIX, and densitometric analysis of blots was plotted. B) The recruitment of CHPMP4B to LdLPVs in these cell lines was evaluated. Representative images of infected cells, labeled with CHMP4B (red), LAMP1 (green), and DAPI (blue), are shown. Also shown are the proportions of LdLPVs shCTRL, TSG101KD and ALIXKD that recruited CHMP4B. At least 100 LdLPVs were measured per coverslip, per time point, and treatment. C) The sizes of LdLPVs in shCTRL, TSG101KD, and ALIXKD were measured and plotted. At least 100 LdLPV sizes were measured per coverslip, per time point, and treatment. D) The number of parasites in infected shCTRL, TSG101KD, and ALIXKD were enumerated. At least 100 macrophages were counted per coverslip, treatment, and time point. A one-way ANOVA was performed, and statistical significance was determined by *post hoc* Tukey's honest significant difference test. For western blotting validation, an unpaired student *T* test was performed for statistical analysis. Data were compiled and graphed using GraphPad Prism 8. *$p < 0.05$, **$p < 0.01$, ***$p < 0.001$, ****$p < 0.0001$. N.S, not significant.
(TIF)

**S7 Fig. A2 expression is sustained in shCTRL, TSG10KD and ALIXKD THP-1 cell lines.** After the differentiation of THP-1 monocytes into macrophages, they were infected with infective metacyclic promastigote forms of *L. donovani*. At 24, 48, and 72 hrs post-infection, cells on coverslips were fixed and processed by immunofluorescence labeling for the detection of A2. At least 100 parasites were counted per coverslip, treatment, and time point. The percentage of parasites expressing A2 was compiled and graphed using GraphPad Prism 8. A one-way ANOVA was performed, and statistical significance was determined by *post hoc* Tukey's honest significant difference test. Data was compiled and graphed using GraphPad Prism 8. *$p < 0.05$, **$p < 0.01$, ***$p < 0.001$, ****$p < 0.0001$. N.S, not significant.
(TIF)

**S8 Fig. Parasites recovered from infection retain their infectivity.** *L. donovani* amastigotes were recovered from shCTRL TSG101KD and ALIXKD THP-1 cell lines after 72 hours of infection. The parasites were counted and then added to THP-1 macrophages on coverslips in 6-well plates at a 5:1 parasite-to-macrophage infection ratio. Infections were scored at 24, 48, and 72 hours of infection. At least 100 macrophages were counted per coverslip in cells infected with recovered parasites. A one-way ANOVA was performed, and statistical significance was determined by *post hoc* Tukey's honest significant difference test. Data was compiled and graphed using GraphPad Prism 8 *$p < 0.05$, **$p < 0.01$, ***$p < 0.001$, ****$p < 0.0001$. N.S, not significant.
(TIF)

**S1 Data. Supplemental data.** Raw data used to generate figures in the manuscript.
(XLSX)

## Author contributions

**Conceptualization:** Peter E. Kima.

**Data curation:** Javier Rosero.

**Investigation:** Javier Rosero, Peter E. Kima.

**Supervision:** Peter E. Kima.

**Writing – original draft:** Peter E. Kima.

**Writing – review & editing:** Javier Rosero, Peter E. Kima.

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
