## [Decision Letter · Decision Letter 0]

29 Apr 2025

A non-canonical activation of the host’s ESCRT machinery is required for the scission of parasitophorous vacuoles and the replication of Leishmaniadonovani

PLOS Pathogens

Dear Dr. Kima,

Thank you for submitting your manuscript to PLOS Pathogens. After careful consideration, we feel that it has merit but does not fully meet PLOS Pathogens's publication criteria as it currently stands. Therefore, we invite you to submit a revised version of the manuscript that addresses the points raised during the review process.

Please submit your revised manuscript within 60 days Jun 28 2025 11:59PM. If you will need more time than this to complete your revisions, please reply to this message or contact the journal office at plospathogens@plos.org. Please include the following items when submitting your revised manuscript:

We look forward to receiving your revised manuscript.

Kind regards,

Tracey J. Lamb

Section Editor

PLOS Pathogens

Tracey Lamb

Section Editor

PLOS Pathogens

Editor-in-Chief

PLOS Pathogens

orcid.org/0000-0003-2946-9497

Michael Malim

Editor-in-Chief

PLOS Pathogens

orcid.org/0000-0002-7699-2064

**Journal Requirements:**

At this stage, the following Authors/Authors require contributions: Javier E Rosero, and Peter E Kima. Please ensure that the full contributions of each author are acknowledged in the "Add/Edit/Remove Authors" section of our submission form.

https://journals.plos.org/plospathogens/s/submission-guidelines#loc-parts-of-a-submission

4) We notice that your supplementary Figures are included in the manuscript file. Please remove them and upload them with the file type 'Supporting Information'. Please ensure that each Supporting Information file has a legend listed in the manuscript after the references list.

Potential Copyright Issues:

i) Figures 1A, 1E, 2A, 4A, 4D, 5A, 5B, 7A, and Graphical Abstract. Please confirm whether you drew the images / clip-art within the figure panels by hand. If you did not draw the images, please provide (a) a link to the source of the images or icons and their license / terms of use; or (b) written permission from the copyright holder to publish the images or icons under our CC BY 4.0 license. Alternatively, you may replace the images with open source alternatives. See these open source resources you may use to replace images / clip-art:

6) We note that your Data Availability Statement is currently as follows: "All relevant data are within the manuscript and its Supporting information files." Please confirm at this time whether or not your submission contains all raw data required to replicate the results of your study. Authors must share the “minimal data set” for their submission. PLOS defines the minimal data set to consist of the data required to replicate all study findings reported in the article, as well as related metadata and methods (https://journals.plos.org/plosone/s/data-availability#loc-minimal-data-set-definition).

7) Please amend your detailed Financial Disclosure statement. This is published with the article. It must therefore be completed in full sentences and contain the exact wording you wish to be published.

8) Your current Financial Disclosure states, "University of Florida Research fund." However, your funding information on the submission form indicates no funds. Please ensure that the funders and grant numbers match between the Financial Disclosure field and the Funding Information tab in your submission form. Note that the funders must be provided in the same order in both places as well.

**Reviewers' Comments:**

Reviewer's Responses to Questions

**Part I - Summary**

Reviewer #1: Rosero and Kima report on the role of the ESCRT machinery in the scission of parasitophorous vacuoles containing the parasite Leishmania donovani (LdLPVs) as well as in the replication of the parasite. They found that ESCRT components are constitutively recruited to LdLPVs, in a PI(3,4)P2-dependent manner. Knockdown of ALIX led to a significant reduction in parasite burden and replication and to an increase in LdLPV size. Overall, this is an interesting study which addresses an important question in the cell biology of intracellular infection, namely how do individual vacuoles replicate at the same time than the pathogen they harbor. However, a number of issues need to be addressed to strenghthen the conclusions of this study.

Reviewer #2: Rosero and Kima provide insight into the role of ESCRT in Leishmania biogenesis. The authors have observed and quantified the recruitment of several of the ESCRT components and then used shRNAs and dominant negative overexpression to manipulate the ESCRT pathway to demonstrate the role of ESCRT in parasite replication. The KD of ALIX prevents the scission of LdLPVs and multiple parasites buildup into the compartment with the absence of LdLPV division.

Reviewer #3: The provided manuscript evaluated the role of ESCRT machinery in Leishmania donovani- infected macrophages. L. donovani invade macrophages, and another immune cells, as host cells, where they occupy a parasitophorous vacuole (PV) to differentiate and replicate. Bacteria that invade host cells have developed secretion system to introduce their proteins into the host cell, but no such system has been found in Leishmania spps. During replication within the host cell, the entire PV is split as well. The authors suggested that the host ESCRT machinery is involved in this process. They evaluated the role of ESCRT I protein TSG101, and ESCRT I accessory protein ALIX, seeing that ALIX was necessary for proper division of the PV after parasite replication. KD of ALIX resulted in PV containing multiple parasites. ESCRT III protein CHMP2B and CHMP4B were implicated in the division, and their recruitment to PV was used as a measure of typical ESCRT pathway function. When CHMP4B recruitment was reduced (because of the ALIX KD), this followed by a reduction in vacuole size, division, and parasite infectivity. I agree with the authors on their conclusions about the non-conical activation of these ESCRT proteins based on the data shown. The authors also implicate PI(3,4)P2 as being the phosphoinositide to play a role in recruiting ESCRT machinery. I don’t think this specific PI, or any PI, can be implicated from this data as presented.

I recommend this paper to be accepted with minor revisions. Clarification of the methodology behind the experiments included in Figure 3 is required to evaluate the conclusions made about PI(3,4)P2. Clarification in the formatting of the figures/images, and the corresponding text will make the conclusions regarding the localization of ESCRT proteins and the role of ALIX clearer to readers. Overall the data and methodology are sound, but some of the descriptions of the images are a bit difficult to follow.

**Part II – Major Issues: Key Experiments Required for Acceptance**

Reviewer #1: 1- An issue that remains to be addressed is why L. donovani fails to replicate in larger vacuoles observed in ALIX knockdown (KD) cells. In addition to affecting vacuole scission, it is possible that ALIX KD alters trafficking to and from the vacuole. It could be that ALIX is required for the acquisition of host factors essential for parasite growth. This issue may be addressed using L. mexicana complex parasites which thrive in spacious vacuoles. The authors should include these parasites to determine whether their replication is affected in ALIX KD cells.

2- Figure 1F, the authors show that overexpression of both CHMP4B and VPS4Adn in RAW cells significantly increased their recruitment to LdLPVs. It would be of interest to compare the growth of L. donovani in control RAW cells and in the CHMP4B/VPS4Adn RAW cells overexpressors.

Reviewer #2: The authors make too strong of an argument for PI(3,4)P2 based on their immunostaining. The antibody (Z-P034) used does have a high enough specificity to rule out PI(4)P for example. Further the authors report in the Materials and Methods section the use of 0.5% Triton X-100 where the manufacture’s protocol warns against exceeding 0.2% Triton X-100 to prevent extraction of lipid populations. While the signal is likely PI(3,4)P2 it could be validated with PI binding proteins fused to GFP which have been well characterized. Further some to the real signal me be extracted with the high Triton X-100 concentration. Please consider mentioning the limitations or directly address the shortcomings with additional experiments.

The work would strongly benefit from electron micrographs (or even super resolution imaging may suffice) of the ALIX-KD to better document the alterations of the parasites and the LdLPVs, however this is not required for publication.

The viability and parasitic health is not well established in the paper; if possible/practical please demonstrate the viability and infectivity of the parasitic progeny from the KD, DN and drug treated conditions.

Reviewer #3: Major Issues:

“The phosphoinositide PI(3,4)P2 on LdLPVs may be the target for recruitment of ESCRT molecules”

I’m confused about the conclusion being drawn here. The title for this section says that PI(3,4)P2 may be the target for ESCRT protein recruitment. PI(3,4)P2 is present on both the vacuoles that surround the latex beads and the Leishmania parasites, meaning it could just always be associated with phagocytosed particles, and not specific to Ld. The authors state they wanted to confirm that PI(3,4)P2 is present on the LdLPV because it has been shown to recruit ESCRT II. Images of infected cells stained with an antibody for PI(3,4)P2, show that it is recruited to LdLPV, and that recruitment is diminished when cells are treated with a general inhibitor of phosphoinositide synthesis (LY294002). They measure decreases CHMP4B. The conclusions is that PI(3,4)P2 is a target of ESCRT, and later in the discussion mention ALIX as the following step in recruitment. I think this is an overstatement, I don’t think this data definitively shows that PI(3,4)P2 is responsible for recruitment of subsequent ESCRT proteins. 1) The reduction in CHMP4B recruitment is not statistically significant (or it’s not stated to be, there is no p value listed so I assume it is not significant, but may have been a mistake). I don’t think the conclusion that PI(3,4)P2 is involved in ESCRT recruitment is supported if that reduction is not significant. 2) The authors haven’t shown that other phosphoinositides aren’t playing a role. The inhibitor used is a broad inhibitor, and there could be may downstream trafficking effects, which is stated in the discussion (long term treatment will kill the parasites by blocking one of their mechanism of evading macrophage activation). Evidence that other phosphoinositidies are not recruited/disrupted, and evidence that the antibody that they’re using is truly specific to PI(3,4)P2 and no other phosphoinositide is needed to draw this conclusion. If all phosphoinositides are disrupted, then even in the absence of PI(3,4)P2, if others are also absent, then they could be responsible for the phenotype. 3) To connect this phosphoinositide to ALIX in the discussion is a stretch, as only CHMP4B recruitment is measured. While I agree that it’s a reasonable hypothesis, there isn’t data presented here to support that conclusion.

There is a more nuanced discussion on this in the discussion, so I believe the authors understand the complexities of these interactions, but I think that nuance isn’t translated in the explanation of the results, or the following sentence from the discussion:

“Our studies revealed a chain of events that commences with the display of the phosphoinositide, PI(3,4)P2 on LdLPVs, followed by the recruitment of ALIX to LdLPVs.”

To summarize, I think the statistical significance of 3B needs to be clarified. Experiments showing that this chemical blocker only effect PI(3,4)P2 recruitment and not any other PI are required, or the language could be modified to not overstate the role of PI(3,4)P2

PI The inhibitor you used can inhibit the synthesis of other phosphoinositides, and therefore other forms in other organelles could be playing a role in ESCRT recruitment. I think this sentence, as written, suggests you had data that showed that ALIX is recruited in response to PI(3,4)P2, however you measured only CHMP4B (ESCRT II) recruitment. While I agree with your later points that ALIX plays a more essential role than TSG101 on the ESCRT I front, and KD of ALIX led to less CHMP4B recruitment, so it’s a reasonable hypothesis, but I think it’s overstating to say that PI(3,4)P2 is followed by ALIX from the data present in this paper.

**Part III – Minor Issues: Editorial and Data Presentation Modifications**

Reviewer #1: 3- Figure 4 and Figure 6 are very similar. Whereas data shown in Fig 4 were obtained with RAW cells, data shown in Fig 6 were obtained with THP-1 cells. Figure 6 may be moved to the Supplemental data section.

4- Bottom of page 8, the authors state that "there was some increase in the frequency of LdLPVs that harbored more than 2 parasites (not shown)". Authors must show the data or remove this sentence.

5- Figure 4D and 4G, the authors state in the text (page 9) that "such larger LdLPVs harbored more that 4-10 parasites". The authors must quantify the number of parasites in the larger vacuoles in ALIX KD cells.

6- The authors must indicate at which time points post-infection were taken the images shown in Figures 1B, D, F, 4D, 5A, 6B.

7- On several figures, font size must be increased to improve legibility.

Reviewer #2: The paper is well written and moves the field forward.

Reviewer #3: Figure formatting: In the way this was submitted, all of the figures were not formatted clearly.

Figure 1 is split across 3 pages in it’s current form.

Stretched/Squished text that is difficult to read: Figure 2 graphs C-F, Figure 4 graphs E-G

Diagrams with small text/stretched that make it difficult to read: Figure 1A and E, Figure 2A, 4 A and D, Figure 5 A and B, and Figure 7A

Also, some of the graphs have titles and some don’t. Where there are titles, I think they’re a little confusing (ex, Fig 2 C-F). Can they be simplified or removed?

The text in Figure 6 seems consistent in sizing and is clear to read, not blurry, and straight. Can be used an example.

For all images: Need scale bars. Each individual fluorophore should be showed in white, and color only used in the merge, especially because the colors switch for LAMP1 depending on the fluorophore of each ESCRT protein. Also, the colors can be changed so they are consistent in merge (LAMP1 can be made red in all of the merges, whether you “collected” it as red or not, as long as all fluorophores are listed)

Introduction

“Leishmania donovani is the causative agent of visceral leishmaniasis that has an estimated 50,000 to 90,000 new cases per year, 95% of which are fatal if left untreated (https://www.who.int/news-room/fact-sheets/detail/leishmaniasis).”

Direct links to webpages should not be cited, the original source material should be cited

“Interestingly, LPVs that harbor L. donovani (LdLPVs) acquire late endosomal characteristics slowly, as evidenced by their prolonged retention of Rab5 among other early endosomal molecules [6]. “

The citation of retention of Rab5 is incorrect, cited paper doesn’t reference Rab5 at all. Also this sentence starts with saying LdLPVs acquire late endosomal characteristics, but then you mention Rab5, an early endosomal marker. So I’m assuming the authors mean to say acquire early endosomal characteristics?

Figure 1

The figure 1A does not match what is said in the text. Also, LAMP1 is written as all uppercase in the text, but all lowercase in the figures, this should be made consistent across the entire paper.

At least 1 outline/zoom in of the LdLPV would be helpful for each image, with arrows pointing to relevant features, (like what is seen in Figure 5). It’s hard to understand is being calling boundaries and periphery. Additionally, the explanation of CHMP2B and 4B forming a “neck” of the horn while the rim (LAMP1) is wider is not clear to me from these images. Looking at the merged picture for CHMP4B infected, in the LPV on the bottom right corner, CHMP4B and LAMP1 seem to have complete overlap of each other where they label together, while in the LPV directly to the left/slightly above the nucleus looks to have CHMP4B in more localized spots compared to LAMP1. I think an outline and arrows would be helpful to more clearly explain your point here, or a diagram, similar to Figure 5. I will say the image from supplemental 1 is much clearer than the image included in figure 1. To me, it seems there is a lot of similar colocalization of LAMP1 and CHMP4B, 2B, and Vps4A even in the uninfected cells, so the latex bead experiment is a great response to that for CHMP2B and 4B. Was this same experiment repeated for TSG101 or Vps4A? If so, the data should be included or mentioned.

The Vps4A data in 1b and 1c is not mentioned at all in the text, and that also looks like the colocalization is the same in infected vs uninfected. The overexpression experiment is clear and the conclusion makes sense, but I think the previous data needs to be mentioned/explained.

For 1F, the text states that a dual transfection of the Vps4 mutant and CHMP2B was used, and then the image and following sentences mention CHMP4B, so I assume 2B is a typo.

For 1g, it is not mentioned in the text. Is this measuring recruitment of both CHMP4B and VPS4? Or some kind of ratio (given that the titles is “CHMP4B/VPS4Adn”)? The figure legend only mentions VPS4.

See notes for “Figure Formatting.” Can the individual points for 1D and 1G be added as in 1C?

Figure 2

Overall this experiment is clear, as is the data. It seems that the CHMP4B is more localized within the LPV, rather than on the periphery as detailed in Figure 1b. How is LLOMe-mediated damage repaired? Does the ESCRT machinery associate and pinch off the damaged membrane to be sent to the plasma membrane? In which case, you’d expect to see CHMP4B on the outside of the PV, not inside. Do you have any explanation for what CHMP4B could be doing, and how it is getting inside the PV? Perhaps more appropriate for the discussion section, but a more detailed of how LLOMe-damage is typically repaired would strengthen the conclusions.

For the figure legend in 1, you list as A) description. B) description. In the figure legend here, you list description (A), description (B). Pick one format and be consistent. Can each individual point be added to 2D and F as seen in Fig 1C?

See notes for “Figure Formatting.”

Figure 3

Why are two different time points used in this experiment (48 hours in no drug versus 24+8 in the treated)? The method section says 48 hours of infection and then 8 hours of treatment, which is not what the figure legend says. If you have images of the vehicle control, this would be a more appropriate comparison.

Most points were already mentioned in “major revisions” but another point to make is the change in PV morphology. It seems that PI is on the leishmania, but doesn’t associate with the entire PV? Or the PV is smaller? If we compare to previous figures looking at how LAMP1 outlined the PV, the PV looks to be a different shape here? An explanation/interpretation of this staining difference would be helpful

On the figure, for Latex beads, no drug, DAPI staining, a star is used, but there isn’t an explanation for what it denotes in the figure legend, nor is there an explanation for the arrows. Two of the labels for “PI(3,4)P2” are missing the comma.

See notes for “Figure Formatting.” Can each individual point be added to 3B as seen in Fig 1C?

Figure 4

The labeling in the text says the western blots for the KD of TSG101 and ALIX are 4B, and selected lines are shown in 2C? But that is not consistent with the actual figure numbering. Overall, these experiments are clear and the data supports the conclusion.

See notes for “Figure Formatting.” Can each individual point be added to 4B-G as seen in Fig 1C?

Figure 5

The images used here, with the arrows and cut outs showing the zoom ins, and the text description, are extremely effective, and this same approach should be taken with earlier images as well to point out the key features, especially in figure 1. But with the colors changed to white for the individual channels and colors only in merge (as stated in the “Figure Formatting” note). This explanation of the results and conclusions in this section are very clear. The diagrams are stretched, and the text is too small to read, but they are effective, so corrected formatting would fix. If diagrams like this could be added to figure 1 and 2, that would really improve the readers ability to follow along.

Can the statistical analysis on part C be added to the figure legend? Also each individual point, as seen in Fig 1C.

Figure 6

It is stated that the number of LdLPVs is fewer in ALIXKD cells, but there is no quantification/measurement presented. I recognize that may be difficult, given the disruption of LAMP1 signal. It seems that the disruption of LAMP1 distribution is more extreme in the THP-1 cells than the RAW macrophages used in Figure 4. Can you comment on that? Otherwise, this data is clear, the conclusions are well supported.

See notes in the “Figure Formatting” Section. Can each individual point be added to each graph as seen in Fig 1C?

Figure 7

Need to add the mention of Fig 7C into the text of the paper (the experiment is mentioned, but not the words “Figure 7C”). This experiment is key to showing the survival of the parasite but lack of PV division, nice experiment.

See notes in the “Figure Formatting” Section.

Discussion

“Together, these results show that ALIX plays a central role in recruiting the ESCRT machinery to LdLPVs where it catalyzes the scission of this of the LdLPV pseudo-organelle. That a non-canonical activation of the ESCRT machinery plays a role in the division of LdLPVs is a new function for the ESCRT machinery.”

I think this conclusion is well supported by the data presented, but there are a few grammatical errors. The additional “of this” in the first sentence. The second sentence is incomplete. Potentially rephrase to “This non-canonical activation of the ESCRT machinery playing a role in the division of LdLPVs is a new function for the ESCRT machinery.”

To repeat my earlier point from figure 3, which perhaps could be included in the discussion rather than the results, how is LLOMe-mediated damage repaired? Does the ESCRT machinery associate and pinch off the damaged membrane to be sent to the plasma membrane? In which case, you’d expect to see CHMP4B on the outside of the PV, not inside. Do you have any explanation for what CHMP4B could be doing, and how it is getting inside the PV?

“Based on our observations, we suggest that ESCRT components are recruited constitutively to LdLPVs and are anchored to phosphoinositides displayed on the LdLPV membrane.”

I agree with this point, and the following explanation in this paragraph. It is a more nuanced explanation of this data than the initial conclusion stated at the start of the discussion. However, I don’t understand this statement:

“The parasite plays a role in synthesizing the phosphoinositide displayed on the LdLPV membrane.”

Which role do you mean? I interpret this to mean that the parasite itself is synthesizing phosphoinositides that are transferred to the PV membrane? Or that parasite kinases are phosphorylating the PIs on the PV membrane? Which hasn’t been addressed in any data in this paper. Please clarify.

You state that the graphical abstract is “highlighting that when ALIX levels are limiting, LdLPVs are large and can harbor 3- 10 parasites”, I don’t think that’s clear in the diagram. Are you using the text size to indicate the amount of ALIX? Maybe you could use [ALIX] or a small up or down arrow?

It’s interesting that the prevention of the PV separation prevents the eventual dispersal of the parasite to new cells. Given that L. amazonensis has a communal PV for multiple parasites, can you elaborate on how you suspect these pathways would be used in L. amazonensis? Would they still require ESCRT machinery? If the parasites still drive expression of ALIX, it could suggest other roles for ALIX, or ESCRT machinery generally, outside of PV division.

PLOS authors have the option to publish the peer review history of their article (what does this mean? ). If published, this will include your full peer review and any attached files.

**Do you want your identity to be public for this peer review?** For information about this choice, including consent withdrawal, please see our Privacy Policy .

Reviewer #1: No

Reviewer #2: No

Reviewer #3: **Yes: ** Sara Fresard

**Figure resubmission:**

**Reproducibility:**



---

## [Decision Letter · Decision Letter 1]

21 Aug 2025

PPATHOGENS-D-25-00079R1

A non-canonical activation of the host’s ESCRT machinery is required for the scission of parasitophorous vacuoles and the replication of Leishmania donovani

PLOS Pathogens

Dear Dr. Kima,

Thank you for submitting your manuscript to PLOS Pathogens. After careful consideration, we feel that it has merit but does not fully meet PLOS Pathogens's publication criteria as it currently stands. Therefore, we invite you to submit a revised version of the manuscript that addresses the points raised during the review process.

Please submit your revised manuscript within 30 days Oct 20 2025 11:59PM. If you will need more time than this to complete your revisions, please reply to this message or contact the journal office at plospathogens@plos.org. Please include the following items when submitting your revised manuscript:

We look forward to receiving your revised manuscript.

Kind regards,

Tracey J. Lamb

Section Editor

PLOS Pathogens

Tracey Lamb

Section Editor

PLOS Pathogens

Sumita Bhaduri-McIntosh

Editor-in-Chief

PLOS Pathogens

orcid.org/0000-0003-2946-9497

Michael Malim

Editor-in-Chief

PLOS Pathogens

orcid.org/0000-0002-7699-2064

**Additional Editor Comments:**

Please address all comments from reviewer 3, particularly with respect to adding additional supplemental figure on how the image analysis was performed and the images scored.

**Journal Requirements:**

1) We note that your Data Availability Statement is currently as follows: "All relevant data are within the manuscript and its Supporting information files.". Please confirm at this time whether or not your submission contains all raw data required to replicate the results of your study. Authors must share the “minimal data set” for their submission. PLOS defines the minimal data set to consist of the data required to replicate all study findings reported in the article, as well as related metadata and methods (https://journals.plos.org/plosone/s/data-availability#loc-minimal-data-set-definition).

**Reviewers' Comments:**

Reviewer's Responses to Questions

**Part I - Summary**

Reviewer #1: The authors have properly addressed my concerns.

Reviewer #2: The authors have a very nice manuscript, and the work moves the field forward. The authors need to be much more specific as to how they score a LdLPV. How is this assessed (see below)? Some of the quantification seems to be from subjective calls of association. There is not a clear description of how you are doing your image analysis. If it is scored by hand and not an automated analysis the parameters being used as criteria for the assessment needs to be very well described.

Reviewer #3: This study evaluates the recruitment of the host cell ESCRT machinery to the Leishmania donovani parasitophorous vacuole. When Ld divide, the parasitophorous vacuole must divide as well, and this study shows compelling evidence of he role of ESCRT machinery, specifically ALIX, playing a role in this division.This study explores an interesting question of ESCRT recruitment, and they mention the implications of this work in other intracellular pathogens. The response of the authors to the reviewers was convincing, and the revised manuscript has simplified the experiments to provide a more streamlined story. The figures are more clear and effective than the first draft. There still remains a few typos that result from the shuffling of figures (ie. the data of 1D is referred to as 1C in a few spots), but overall this paper should be accepted.

**Part II – Major Issues: Key Experiments Required for Acceptance**

Reviewer #1: (No Response)

Reviewer #2: Fig 1B Pearson's correlation coefficients should be reported for colocalization with LAMP1

Fig 1C The Axis should be labeled as LAMP1 since not every structure is parasitophorous vacuole. In all figures the axis label should be more definitive. It is not clear how you define structures.

Fig 1D In this example cell there is only a single phagosome. In the Leishmania vacuoles (not shown here) there are more units per cell than the latex beads, is this considered in the quantification? Are you using all of the LAMP1 signal as your denominator? Basically, there is not enough clarity on the scoring metrics used. It would only be fair to quantify the LAMP1 signal within a certain distance of the bead or Leishmania, and then to score the CHMP2B/4B on that compartment.

Fig1F Again Pearson’s CC should be done and on both infected and non-infected.

Collectively there is an issue with parsing out what is a vacuole verses a late endosome / lysosome. Please note that late endosomes and lysosomes have LAMP1 signal. Using the Leishmania DAPI signal the compartments can be scored. It would be ideal if in a supplemental figure a detailed analysis pipeline can be outlined with sample images highlighted to show the reader what is being scored. Is LAMP1 being scored as an LdLPV or do you require the DAPI signal within and LAMP1 compartment?

There is not very much said about the reticular morphology of the LAMP1 signal. Please add some notes on this in the early section of the results and comment on this morphology shift in the discussion. Is the LdLPV essentially the same as the Salmonella containing compartment you mention in the introduction? Please add a comment on this in the discussion.

Reviewer #3: (No Response)

**Part III – Minor Issues: Editorial and Data Presentation Modifications**

Reviewer #1: (No Response)

Reviewer #2: Fig 6B If you look closely at the LAMP1 (green) in the CHMP4B in shCTRL there is a very scant red signal. Is this an artifact of image prep or was the image color separated in Photoshop? Please make sure the green is all green. You will have to zoom in and look carefully, but there is red there.

Reviewer #3: (No Response)

PLOS authors have the option to publish the peer review history of their article (what does this mean? ). If published, this will include your full peer review and any attached files.

**Do you want your identity to be public for this peer review?** For information about this choice, including consent withdrawal, please see our Privacy Policy .

Reviewer #1: No

Reviewer #2: No

Reviewer #3: **Yes: ** Sara Fresard

**Figure resubmission:**
---

## [Editor Report · Decision Letter 2]

2 Sep 2025

Dear Dr. Kima,

We are pleased to inform you that your manuscript 'A non-canonical activation of the host’s ESCRT machinery is required for the scission of parasitophorous vacuoles and the replication of Leishmania donovani' has been provisionally accepted for publication in PLOS Pathogens.

Best regards,

Tracey J. Lamb

Section Editor

PLOS Pathogens

Tracey Lamb

Section Editor

PLOS Pathogens

Sumita Bhaduri-McIntosh

Editor-in-Chief

PLOS Pathogens

orcid.org/0000-0003-2946-9497

Michael Malim

Editor-in-Chief

PLOS Pathogens

orcid.org/0000-0002-7699-2064
---

## [Editor Report · Acceptance letter]

Dear Dr. Kima,

We are delighted to inform you that your manuscript, "A non-canonical activation of the host’s ESCRT machinery is required for the scission of parasitophorous vacuoles and the replication of Leishmania donovani," has been formally accepted for publication in PLOS Pathogens.

Best regards,

Sumita Bhaduri-McIntosh

Editor-in-Chief

PLOS Pathogens

orcid.org/0000-0003-2946-9497

Michael Malim

Editor-in-Chief

PLOS Pathogens

orcid.org/0000-0002-7699-2064